# Interplay between *Pitx2* and *Pax7* temporally governs specification of extraocular muscle stem cells

**Mao Kuriki**, **Amaury Korb**, **Glenda Comai** *, **Shahragim Tajbakhsh** *

Institut Pasteur, Université Paris Cité, CNRS UMR 3738, Stem Cells & Development Unit, Institut Pasteur, Paris, France

* comai@pasteur.fr (GC); shahragim.tajbakhsh@pasteur.fr (ST)

## Abstract

Gene regulatory networks that act upstream of skeletal muscle fate determinants are distinct in different anatomical locations. Despite recent efforts, a clear understanding of the cascade of events underlying the emergence and maintenance of the stem cell pool in specific muscle groups remains unresolved and debated. Here, we invalidated *Pitx2* with multiple *Cre*-driver mice prenatally, postnatally, and during lineage progression. We showed that this gene becomes progressively dispensable for specification and maintenance of the muscle stem (MuSC) cell pool in extraocular muscles (EOMs) despite being, together with Myf5, a major upstream regulator during early development. Moreover, constitutive inactivation of *Pax7* postnatally led to a greater loss of MuSCs in the EOMs compared to the limb. Thus, we propose a relay between *Pitx2*, *Myf5* and *Pax7* for EOM stem cell maintenance. We demonstrate also that MuSCs in the EOMs adopt a quiescent state earlier that those in limb muscles and do not spontaneously proliferate in the adult, yet EOMs have a significantly higher content of Pax7+ MuSCs per area pre- and post-natally. Finally, while limb MuSCs proliferate in the *mdx* mouse model for Duchenne muscular dystrophy, significantly less MuSCs were present in the EOMs of the *mdx* mouse model compared to controls, and they were not proliferative. Overall, our study provides a comprehensive *in vivo* characterisation of MuSC heterogeneity along the body axis and brings further insights into the unusual sparing of EOMs during muscular dystrophy.

## Author summary

Skeletal myogenesis serves as a paradigm to study the mechanisms controlling stem cell commitment, proliferation, and differentiation. Understanding these mechanisms is crucial for addressing their dysregulation in various diseases and ageing. Although skeletal muscle is found throughout the body, differences in embryological origins, regenerative potential, and susceptibility to myopathies exist between head and trunk muscles. Notably, extraocular muscles (EOMs) demonstrate unique resilience in conditions like Duchenne muscular dystrophy. Most research on muscle stem cell (MuSC) biology has focused on limb muscles given their accessibility for experimentation. While the

**Data Availability Statement:** All data are in the manuscript and/or supporting information files.

**Funding:** We acknowledge funding support from the Institut Pasteur, Centre National de la Recherche Scientifique, Agence Nationale de la

Recerche (Laboratoire d'Exellence Revive, Investissement d'Avenir; ANR-10-LABX-73 to ST and ANR-21-CE13-0005 MUSE to GC), Association Française contre les Myopathies (Grant #20510 to ST and #23201 to GC) and the European Research Council (Advanced Grant #101055234 to ST). M.K. was supported by a Post-Doctoral Fellowship from Uehara Memorial Foundation and The Osamu Hayaishi Memorial Scholarship for Study Abroad. The funders did not play any role in the study design, data collection and analysis, decision to publish, or preparation of the manuscript.

transcription factor Pitx2 plays a key role in EOM formation during embryonic development, the mechanisms governing the emergence and maintenance of MuSCs at this location remain unclear. Here, we demonstrate that MuSCs in the EOMs enter a quiescent state earlier than those in limb muscles and do not spontaneously proliferate in adulthood, challenging previous assumptions. Additionally, through genetic analyses, we elucidate the interplay between Pitx2, the MuSC marker Pax7, and the myogenic factor Myf5 in regulating extraocular MuSCs, thus refining our understanding of the genetic cascades that govern craniofacial myogenesis.

## Introduction

Muscle stem cells (MuSCs) arise during late foetal development and contribute to the postnatal growth of muscle fibres. They are the main players for reconstituting muscle fibres during regeneration. This population of stem cells emerges in an asynchronous manner in different anatomical locations, and it exhibits an unusual diversity in developmental origins and genetic regulation depending on the muscle group in which it resides [1–3]. Understanding the physiological mechanisms controlling the development of different muscle groups, and how they are selectively deregulated or spared during disease and ageing, is crucial for devising therapeutic strategies for a wide range of myopathies. Extraocular muscles (EOMs) include 4 recti muscles (superior, inferior, medial, and lateral) and 2 oblique muscles (superior and inferior) for movement of the eyeball, and arise from different cranial mesoderm populations [4–6]. They are spared during ageing and in Duchenne, Becker, and some limb girdle muscular dystrophies [7–9]. While the underlying causes for their sparing are unclear, intrinsic differences in MuSCs of the EOMs have been proposed as contributing factors [10–12]. Yet, the precise mechanisms governing the emergence and maintenance of these and other cranial MuSC populations remain obscure.

Myogenic commitment and differentiation throughout the body is regulated by basic helix-loop-helix myogenic regulatory factors (MRFs) comprising Myf5, Mrf4, Myod and Myogenin (Myog). Mouse knockout models have established a genetic hierarchy where *Myf5*, *Mrf4* and *Myod* control commitment and proliferation of myogenic progenitors, and *Myod*, *Mrf4* and *Myog* are involved in terminal differentiation [1,13,14]. Despite this uniformity in the acquisition of myogenic cell fate, an underlying heterogeneity characterises the embryonic cells that emerge in different anatomical locations. In the trunk, the paralogous transcription factors *Pax3* (Paired Box 3) and *Pax7* (Paired Box 7) regulate the emergence of muscle stem (Pax3+ or Pax7+) and committed (Myf5+, Mrf4+ or Myod+) cells [1,15,16]. Somite-derived myogenic cells are eventually lost by apoptosis from mid-embryogenesis in *Pax3;Pax7* double-mutant mice [17]. In contrast, craniofacial muscles such as EOMs, facial, mastication and subsets of neck muscles, are derived from cranial mesoderm and do not express *Pax3* [2,6,18,19]. The early head mesoderm harbors instead a complementary set of upstream markers including *Tbx1*, *Pitx2* and the *Six* gene family [20–23]. Notably, *Tbx1* (T-Box Transcription Factor 1), is an upstream regulator for the facial, jaw, neck, laryngeal and esophagus muscles, but not of the EOMs [18,20,21,24–26]. Instead, *Pitx2* (Paired Like Homeodomain 2), which is also expressed in head muscles, plays a critical role as an upstream regulator of EOM development, as these muscles are lost in *Pitx2* null mice [27,28], with phenotypes being dependent on the gene dosage [29] and timing of deletion [30].

In contrast to the diversity in upstream regulators, *Pax7* marks all adult MuSCs and their ancestors from mid-embryogenesis [1]. In the trunk, Pax7+ myogenic cells are primed by

prior *Pax3* expression during embryonic development [31]. However, the mechanisms that govern the emergence and maintenance of cranial myogenic cells are poorly understood. As *Pax3* is absent in the EOMs, and *Pax7* is expressed after the MRFs [32,33], it is possible that *Pitx2* or other transcription factors such as the *Six* family members [22,34] specify the Pax7 + population in this location. Yet, as no MuSC lineage-specific deletion of *Pitx2* has been performed in EOMs, it is unclear: i) to what extent *Pitx2* is required temporally for emergence and maintenance of the MuSC population, and its relative function compared to the MRFs and Pax7; ii) if *Pitx2* is continuously required for maintenance of the adult MuSC population. Further, global deletion of *Pax7* in trunk and limb muscles leads to an extensive loss of MuSCs during early postnatal development [35–39]. Deletion of *Pax7* in adult MuSCs results in reduced MuSC numbers and impaired regeneration following muscle injury [40,41]. While *Pax7* is dispensable for EOM formation during embryonic development [30], the presence of MuSCs in the EOMs of postnatal *Pax7* null mice and the relative roles of *Pax7* and *Pitx2* in maintaining these MuSCs remain unresolved. Finally, several reports showed that expression of *Pitx2* in postnatal MuSCs is higher in EOMs than in limb muscles [10,11,42]. As high levels of *Pitx2* are retained in dystrophic and ageing mouse EOMs [42,43] and overexpression of *Pitx2* was reported to enhance the regenerative potential of dystrophic skeletal MuSCs [44], *Pitx2* could potentially contribute to the sparing of EOMs in muscular dystrophies. However, the functional role of *Pitx2* in EOM MuSCs in *mdx* mice has not been formally evaluated to-date.

In this study, we examined the spatiotemporal functional roles of *Pitx2* and *Pax7* in the emergence and maintenance of the EOM progenitor population using combinations of genetically modified and lineage traced mice. Our study provides a qualitative and quantitative analysis of myogenesis from embryonic to postnatal stages *in vivo* and identifies differential cycling states, dynamics of commitment, and establishment of MuSC quiescence in EOMs compared to limb muscles. We show that *Pitx2* is critically required for establishing the EOMs prior to *Pax7* expression but is subsequently dispensable. Moreover, our findings clarify the behavior of MuSCs in the EOMs in *mdx* mice thus providing valuable developmental and clinical insights.

## Results

### Differential temporal dynamics of EOM and limb myogenic cells during development and postnatally

Previous studies suggested that EOMs have more MuSCs per area and per tissue weight than limb muscles such as the *Tibialis anterior* (TA) [45, 46]. To determine if the higher density of MuSCs *in vivo* results from a longer proliferative phase of EOM Pax7+ cells during embryonic and postnatal development, we performed co-immunostaining of Pax7 and the proliferation marker Ki67 on tissues sections from E (embryonic day) 14.5 to 4 months postnatally (Fig 1A and 1B). As expected, the total numbers of Ki67+Pax7+ cells were progressively reduced during postnatal growth in both muscles. Yet, MuSCs in the EOMs stopped proliferating by P (postnatal day) 14, whereas proliferating Pax7+ MuSCs continued to be observed in TA muscle at P20 (Fig 1C). Thus, EOM Pax7+ cells exited the cell cycle earlier than those in the TA. In the TA, cycling Pax7+Ki67+ cells persisted in pre-pubertal muscle (P28) before quiescence was achieved between 7–8 weeks [47] (Fig 1C). Notably, the density of EOM Pax7+ MuSCs remained higher in the orbital layer, probably due to the smaller fibre size compared to the global layer (S1A Fig). Finally, given that lineage tracing experiments suggested that constant myonuclear turnover takes place in the EOMs [45,48], we evaluated the proliferative status of Pax7+ cells in adult EOMs by using 5-ethynyl-2′-deoxyuridine (EdU) uptake (2 weeks;

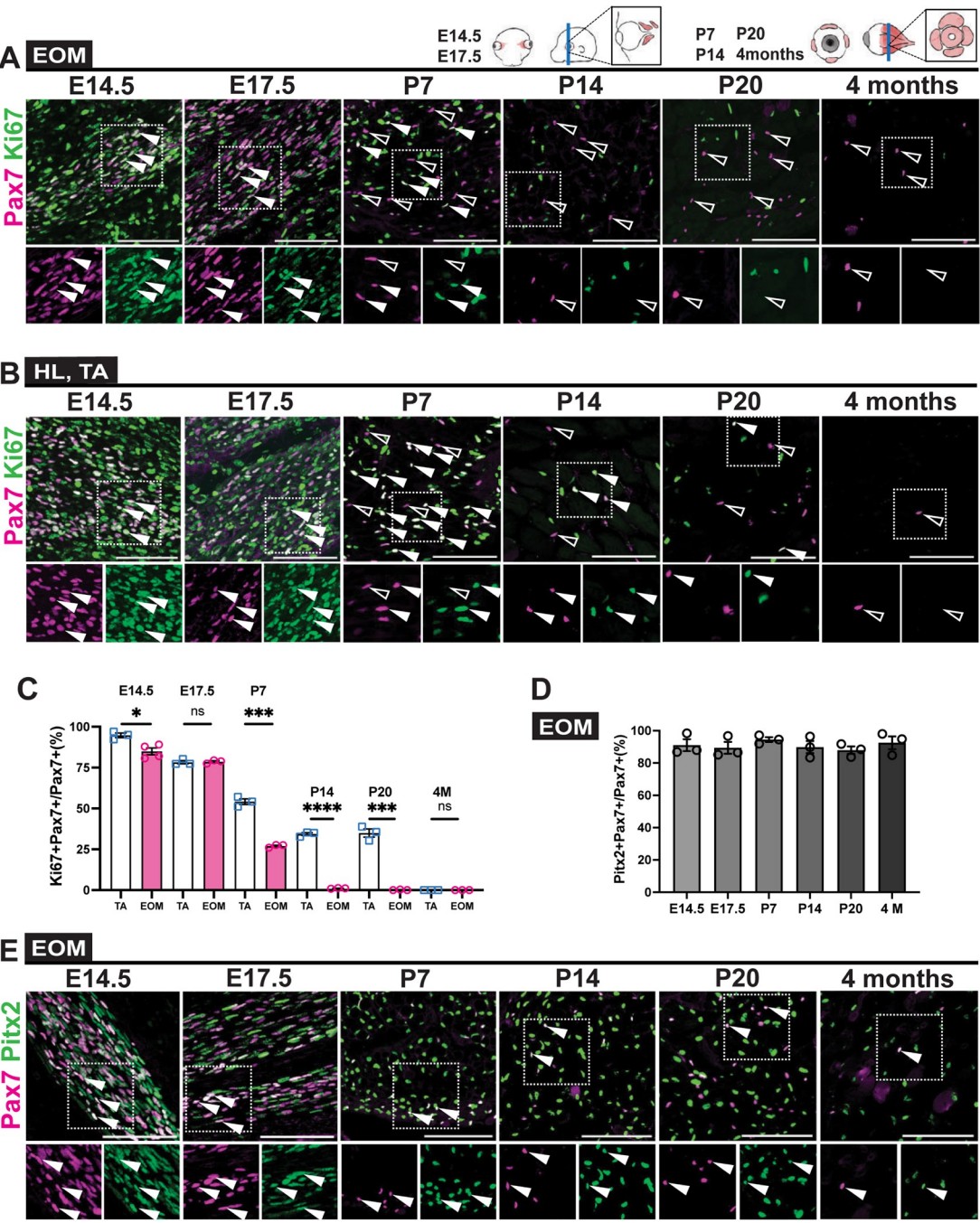

**Fig 1. Postnatal extraocular myogenic cells exit the cell cycle earlier than those in TA muscle. (A,B)** Immunostaining for Pax7 (magenta) and Ki67 (green) at indicated stages from E14.5 to 4 months (4M) at the EOM, HL (E14.5, E17.5) or TA level (P7, 4 months). White arrowheads point to Ki67+Pax7+ cells, open arrowheads to Ki67-Pax7+ cells. Bottom panels, higher magnification views of the area delimited with dots. **(C)** Percentage of Ki67+Pax7+ cells in extraocular, HL and TA muscles at each stage (n = 3 per stage). **(D)** Percentage of Pitx2+Pax7+ cells over total Pax7+ stem cells at indicated stages from E14.5 to 4 months (4M) (n = 3 per stage). **(E)** Immunostaining for Pax7 (magenta) and Pitx2 (green) on EOM sections at indicated stages. White arrowheads point to Pitx2+Pax7+ cells. Bottom panels, higher magnification views of the area delimited with dots. Scale bars: 100μm (A), (B) and (E). Error bars represent mean ± SEM. Two-tailed unpaired Student's t-test. ns, non-significant, P>0.05, *P<0.05, ***P<0.001, ****P<0.0001. EOM, extraocular muscle; HL, hindlimb; TA, Tibialis anterior. All recti EOMs were assessed.

S1B Fig). Accordingly, only a few Pax7+EdU+ were detected in the EOMs, whereas 11.4% double positive cells were detected in the TA (S1C and S1D Fig). Altogether, these data show that the relatively higher number of Pax7+ cells per area in the EOMs cannot be accounted for by a longer or constant proliferative phase. Of note, as *Pitx2* was expressed in the majority of Pax7+ cells in EOMs throughout all stages examined (80–90%; Fig 1D and 1E), Pitx2 alone does not appear to promote proliferation of MuSCs in EOMs in the adult.

## *Pax7* is critical for maintenance of the MuSC pool in EOMs postnatally

To investigate the role of *Pax7* in MuSCs in EOMs compared to limb muscles, we examined $Pax7^{nGFP/nGFP}$ knock-out mice (*Pax7* KO) (Fig 2A and 2B) where *Pax7*-null and heterozygous cells are GFP+ and GFP expression persists in more differentiated cells [21,49]. At P20, the number of GFP+ cells (per area and per 100 fibres) was significantly reduced both in EOMs and TA of *Pax7* KO mice compared to heterozygous controls, although to a greater extent in the EOMs (Fig 2C, S2A–S2C Fig). No differences were depicted among individual recti muscles. Notably, GFP+ cells were mainly lost in the global (inner) EOM layer of *Pax7* mutants, yet this was relatively unchanged in the orbital (outer) layer, thereby pointing to location specific requirements for *Pax7* (S2A and S2B Fig).

Next, we addressed the mechanisms underlying MuSC loss in *Pax7* null EOMs. A greater propensity to differentiate was reported for postnatal *Pax7* mutant MuSCs [39–41]. We performed immunostaining for Myod (commitment marker) and Myogenin (Myog, differentiation marker) in EOM and TA muscles of *Pax7* KO and control mice (Fig 2D and 2E). At P20, a significantly higher fraction of Myod+ and Myogenin+ (MM+) cells in the GFP+ population was observed in *Pax7* KO EOMs (30% in KO vs 5.8% in control; Fig 2F), indicating depletion of the stem cell population by commitment to differentiation. This difference was not notable for the TA at P20. We then examined EOMs and TA of E14.5 *Pax7* KO embryos to assess the role of *Pax7* in maintenance of myogenic progenitors in the foetus (S2D–S2L Fig). At E14.5, the number of Pax7+ cells per area was similar between EOMs and TA of *Pax7* KO embryos (S2D–S2F Fig). However, the fraction of MM+GFP+ cells was already higher in the EOMs of *Pax7* KO embryos compared to controls (S2G–S2I Fig). Notably though, the control TA had ~9-fold and ~2-fold more MM+ GFP+ population compared to the EOMs at P20 (Fig 2F) and E14.5 (S2I Fig), respectively, indicating a more rapid transition from stem to committed and differentiated cells *in vivo* in normal conditions. Evaluation of the expression of Calcitonin receptor (CalcR), which is a marker of postnatal quiescent MuSCs [50], showed that >90% of GFP+ cells in the *Pax7* heterozygous were positive for this marker at P20 as expected, whereas all GFP+ cells in the *Pax7* KO were CalcR-negative in both TA and EOMs, suggesting that in both muscles this residual GFP+ population is perturbed (Fig 2G–2I).

Given that *Pitx2* is a major upstream regulator of the EOM lineage, we assessed whether Pitx2+ cells would expand in the *Pax7* KO. Co-immunostaining for Pax7 and Pitx2 in *Pax7* KO mice at embryonic and adult stages showed that the number of Pitx2+GFP+ cells were similar in the *Pax7* KO and control mice for both EOMs and TA (Figs 2J–2L and S2J–S2L). Moreover, unlike other skeletal muscles, EOMs were reported to contain another population of interstitial stem cells that are Pax7-Pitx2+ [43]. Examination of this interstitial population in mutant versus control EOMs showed similar proportions between both (S2M Fig). Overall, these results suggest a lack of compensation by Pitx2 in *Pax7* KO mice.

Finally, progressive loss of limb and trunk MuSCs in newborn constitutive *Pax7* mutants was reported to be due to a proliferation defect and cell death [37,38]. Cleaved-caspase 3 staining showed higher numbers of dying cells per area in the EOMs of *Pax7* KO compared to control at P1, whereas no significant differences were observed at embryonic or late postnatal

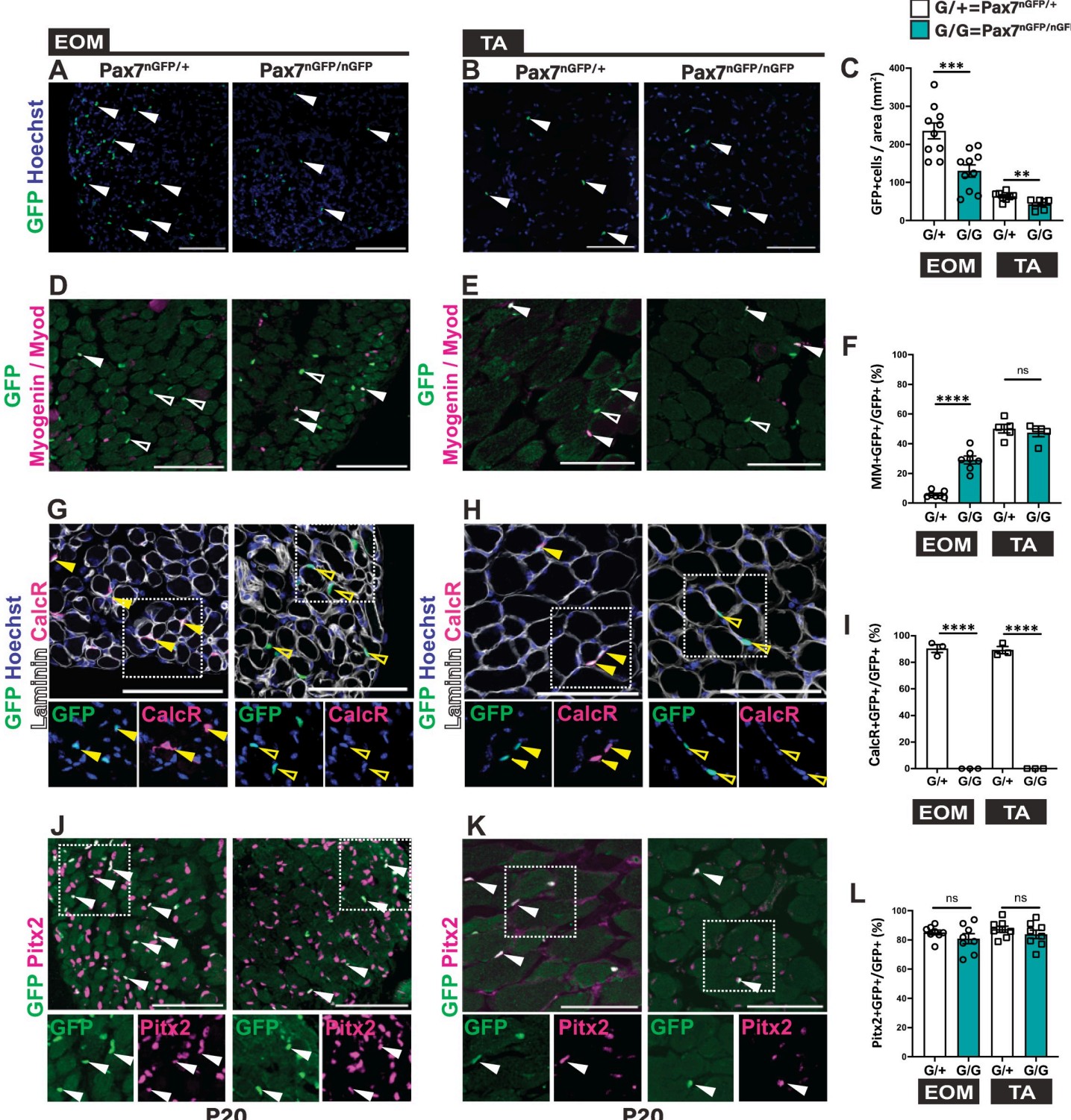

**Fig 2. *Pax7* is critical for the maintenance of the extraocular MuSC pool postnatally. (A, B)** Immunostaining for GFP on EOM and TA muscle sections from *Pax7nGFP/+* and *Pax7nGFP/nGFP* P20 mice. White arrowheads indicate GFP+ cells (green). **(C)** Number of GFP+ cells per area from the immunostaining in (A, B). (EOM n = 10, TA n = 8). **(D, E)** Immunostaining for GFP (green), together with Myod and Myogenin (MM) (magenta) on EOM and TA muscle sections from *Pax7nGFP/+* and *Pax7nGFP/nGFP* P20 mice. White arrowheads indicate MM+GFP+ cells; open arrowheads MM-GFP+ cells. **(F)** Percentage of MM+GFP+ cells over total GFP+ cells from immunostaining in (D,E) (EOM n = 7, TA n = 5). **(G,H)** Immunostaining for GFP (green), Calcitonin receptor (CalcR) (magenta) and Laminin (white) on EOM and TA muscle sections from *Pax7nGFP/+* and *Pax7nGFP/nGFP* P20 mice. Yellow arrowheads indicate CalcR+GFP+ cells; yellow open arrowheads indicate CalcR-GFP+ cells. Bottom

panels, higher magnification views of the area delimited with dots. **(I)** Percentage of CalcR+GFP+ cells over total GFP+ cells from immunostaining in (G,H) (n = 3 each). **(J,K)** Immunostaining for GFP (green), together with Pitx2 (magenta) on EOM and TA muscle sections from $Pax7^{nGFP/+}$ and $Pax7^{nGFP/nGFP}$ P20 mice. White arrowheads indicate Pitx2+GFP+ cells. Bottom panels, higher magnification views of the area delimited with dots. **(L)** Percentage of Pitx2+GFP+ cells over total GFP+ cells from immunostaining in (J, K) (EOM n = 8, TA n = 8). G/+: $Pax7^{nGFP/+}$; G/G: $Pax7^{nGFP/nGFP}$. Scale bars: 200μm (A) and (B), 100μm (D,E,G,H,J,K). Error bars represent mean ±SEM. Two-tailed unpaired Student's t-test. ns, non-significant, P>0.05, **P<0.01, ***P<0.001, ****P<0.0001. EOM, extraocular muscle; TA, Tibialis anterior. All recti EOMs were assessed.

stages (S2N–S2T Fig). Altogether, we conclude that *Pax7* is required for maintenance of the MuSCs in the EOMs, and that invalidation of this gene results in a more significant loss of the upstream population by cell death and more commitment to differentiation, compared to the limb.

## Deletion of *Pitx2* in Myf5+ cells impairs establishment of EOMs and emergence of Pax7+ cells

Cranial muscles are specified temporally after those in the trunk and Pax7+ stem cells appear at later stages during development, after expression of the myogenic markers Myf5 and to a lesser extent Myod [32,33, 51] (Fig 3A). However, it is unclear if expression of the MRFs is a prerequisite for the emergence of the Pax7+ cells themselves.

Therefore, we performed whole mount immunofluorescence (WMIF) analysis on $Myf5^{nlacZ/+}$ mice, where β-galactosidase serves as a proxy for *Myf5* expression [52] (Fig 3A–3C). In the EOM primordium, *Pitx2* is expressed broadly in the periocular mesenchyme and connective tissue cells [53,54]. WMIF showed that most Myf5+ cells expressed *Pitx2* in the EOM anlage throughout the stages analysed (Figs 3B and S3A). In addition, we observed Pax7+ expression from E11.75, only in Myf5+ cells (Fig 3C). From E11.75, EOM Pax7+ cells then rapidly increased in numbers together with Pax7 protein levels (Fig 3B and 3C).

To examine further the emergence of MuSCs in EOMs, we performed lineage tracing of Pax7+ cells originating from Myf5+ cells using the $Myf5^{Cre}$ [55] together with the $R26^{mTmG}$ and $Myf5^{nlacZ}$ reporter mice (Fig 3D). This genetic strategy allowed us to first assess the number of Pax7+ cells with a history of *Myf5*-expression (membrane GFP+), then to detect with high sensitivity, contemporary *Myf5*-expressing cells (β-gal+) that might not have activated *Cre*-expression. Consequently, a broader view on the Myf5+ population, giving rise to and residing within *Pax7*-expressing cells, could be resolved. Remarkably, this analysis revealed that at E11.5 and E14.5, all Pax7+ cells showed a history of *Myf5* expression (Fig 3E).

Based on the dynamics of expression of these transcription factors, we used $Myf5^{Cre}$ to delete *Pitx2* before the onset of *Pax7* expression. $Myf5^{Cre}; Pitx2^{flox/flox}$ E14.5 embryos showed no Pax7+ cells, only a few MM cells, and no myofibres (Fig 3F and 3G). Analysis at foetal stages confirmed the lack of EOMs in the mutant, albeit a small cluster of Pax7+ cells and myofibres were detected in dorsal sections (S3B–S3D Fig). As these EOM remnants contained Pitx2+Pax7+ and Pitx2-Pax7+ cells and correspond to the site of attachment onto the base of the skull, it is possible that $Myf5^{Cre}$ was less efficient in this location or they were derived from Mrf4+ cells [21].

Given that *Myf5* is expressed briefly in muscle progenitors and downregulated when myoblasts commit to differentiation [1], we used the more downstream commitment marker *Myod* to abolish *Pitx2* expression in myoblasts using $Myod^{iCre}$ [56]. Myogenesis appeared unperturbed in the EOMs in this case (Fig 4A–4D). Notably, in stark contrast to the $Myf5^{Cre}$ data, similar numbers of *Pax7*-expressing cells were present in controls and $Myod^{iCre};Pitx2^{flox/flox}$ foetuses (Fig 4B), despite recombination taking place in >90% of the cells as per $Myod^{iCre};R26^{Tomato}$ lineage tracing (S4A and S4B Fig). Taken together, these results show that *Pitx2* is required in Myf5

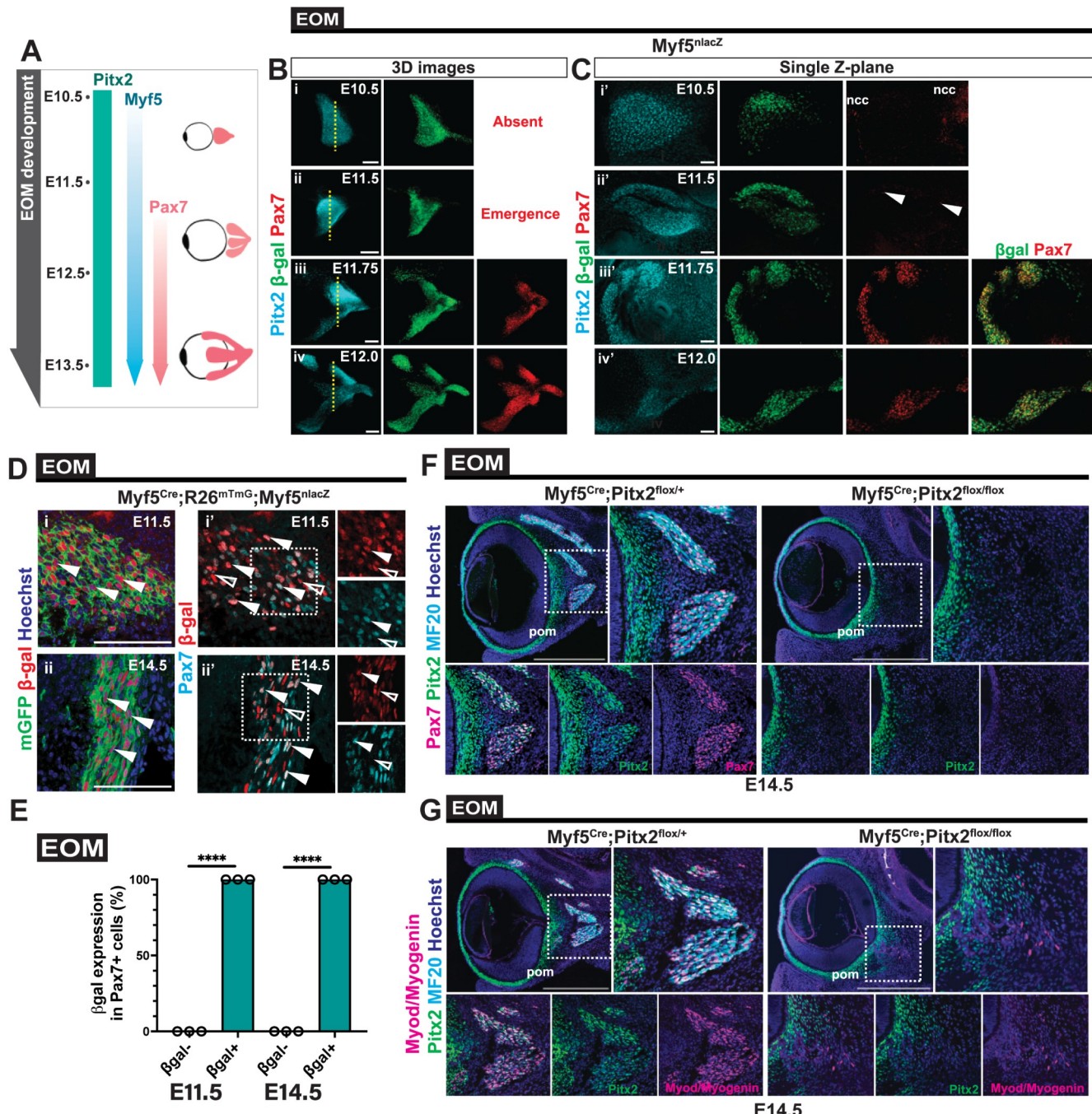

**Fig 3. Deletion of *Pitx2* in *Myf*5-positive cells impairs establishment of EOMs. (A)** Scheme summarising the timing of expression of *Pitx2*, *Myf5* and *Pax7* during EOM development from the anlage stage to individual muscles. **(B)** Whole-mount immunostaining of the EOM anlage of *Myf5^nlacZ* embryos at (i) E10.5, (ii) E11.5, (iii) E11.75, (iv) E12.0 for Pax7 (red), β-gal (green) and Pitx2 (cyan). EOMs were segmented from adjacent head structures and 3D-reconstructed in Imaris. ncc, neural crest cells. Only few myogenic cells express Pax7 at E11.5 (white arrowheads). **(D)** Immunostaining of EOM sections from *Myf5^Cre;R26^mTmG: Myf5^nlacZ* embryos at E11.5 (i, i') and E14.5 (ii, ii') for Pax7 (cyan), β-gal (red) and mGFP (green). White arrowheads in (i) and (ii) indicate mGFP+β-gal+ cells. White arrowheads in (i') and (ii') indicate Pax7+β-gal+ cells. White open arrowheads point to Pax7-β-gal+ cells. Right panels, higher magnification views of the area delimited with dots. **(E)** Percentage of β-gal+ and negative cells over total Pax7+ population in EOMs at E11.5 and E14.5 from the immunostaining in (D) (n = 3 each). **(F, G)** Immunostaining of E14.5 EOM sections from *Myf5^Cre;Pitx2^flox/+* (control) and *Myf5^Cre;Pitx2^flox/flox* (KO) for Pax7 (magenta), Pitx2 (green) and MF20 (cyan) (F) and Myod and Myogenin (MM) (magenta), Pitx2 (green) and MF20 (cyan) (G). Bottom panels, higher magnification views of the area delimited with dots. Periocular mesenchyme (pom). Scale bars: 100μm (B,D), 40μm (C,i'-ii'), 50μm (C,iii'-iv'), 500μm (F,G). Error bars represent mean ± SEM. Two-tailed unpaired Student's t-test. ****P<0.0001. EOM, extraocular muscle. All recti EOMs were assessed.

+ myogenic progenitors for the specification and emergence of the Pax7+ population and is then dispensable upon activation of *Myod*.

Finally, to assess the role of Pax7+ cells postnatally, we ablated *Pax7*-expressing cells and their descendants using a diphteria toxin (DTA) approach (Fig 5A). Induction of recombination at P2 and P3 in *Pax7^{CreERT2};R26^{DTA}* mice [57,58] resulted in complete ablation of the Pax7+ population, the loss of CalcR+ cells, and a significant loss of MM+ cells in both the extraocular and TA muscles when analysed 5 days after the injections (Fig 5B–5J). Moreover, there was no significant increase in the Pitx2+ population (S5A–S5D Fig). This result demonstrates that an independent reservoir of myogenic cells does not compensate for the loss of the Pax7+ myogenic population.

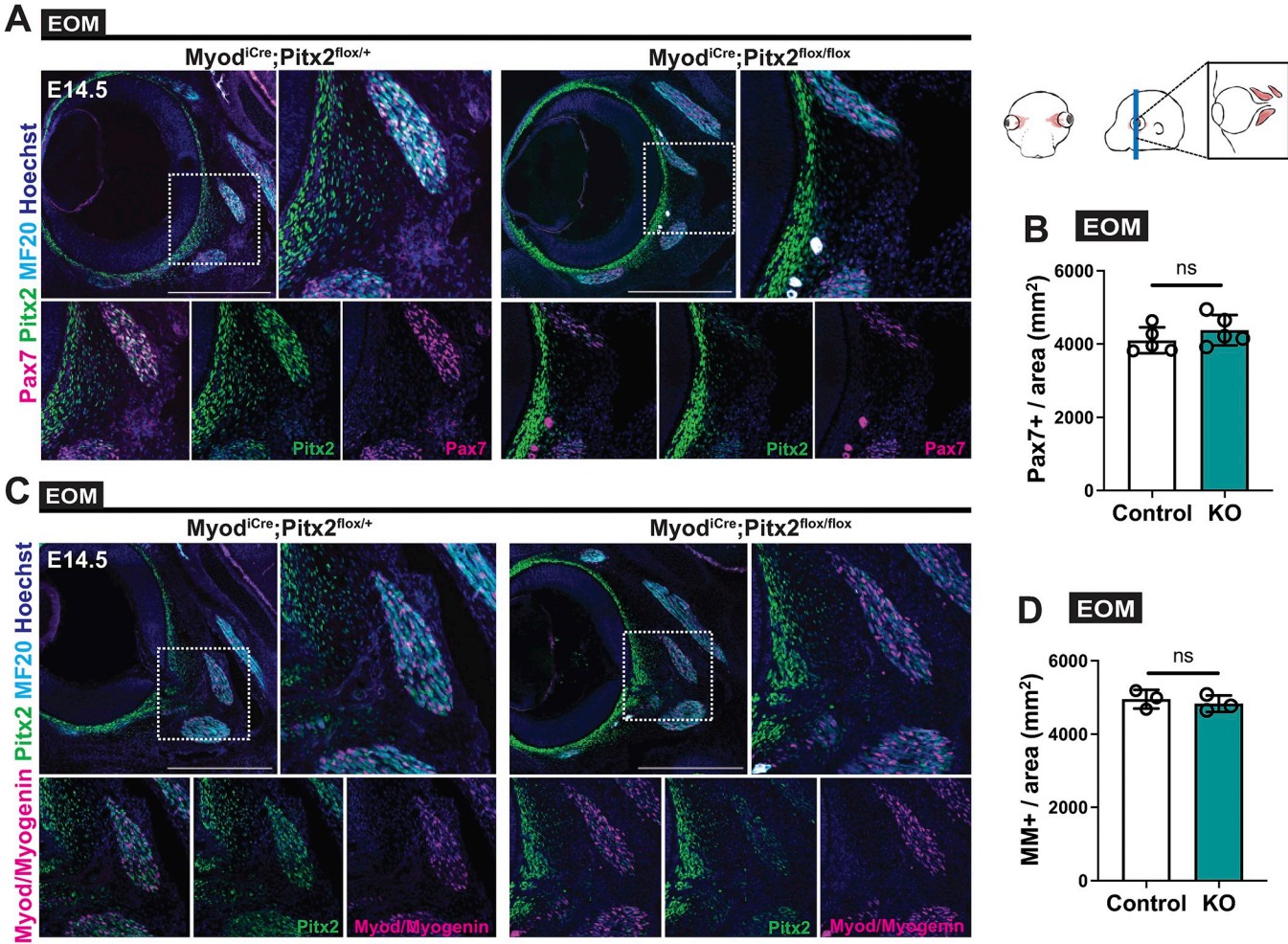

**Fig 4. Normal EOM formation following deletion of *Pitx2* in *Myod*-expressing cells. (A)** Immunostaining of E14.5 EOM sections from *Myod^{iCre};Pitx2^{flox/+}* (control) and *Myod^{iCre};Pitx2^{flox/flox}* (KO) for Pax7 (magenta), Pitx2 (green) and MF20 (cyan). Bottom panels, higher magnification views of the area delimited with dots. **(B)** Number of Pax7+ cells per area in EOM sections from the immunostaining in (A) (n = 5 each). **(C)** Immunostaining of E14.5 EOM sections from *Myod^{iCre};Pitx2^{flox/+}* (control) and *Myod^{iCre};Pitx2^{flox/flox}* (KO) for Myod and Myogenin (MM) (magenta), Pitx2 (green) and MF20 (cyan). Bottom panels, higher magnification views of the area delimited with dots. **(D)** Number of MM+ cells per area in EOM sections from immunostaining in (C) (n = 3). Scale bars: 500μm (A) and (B). Error bars represent mean ± SEM. Two-tailed unpaired Student's t-test. ns, non-significant, P>0.05. EOM, extraocular muscle. All recti EOMs were assessed.

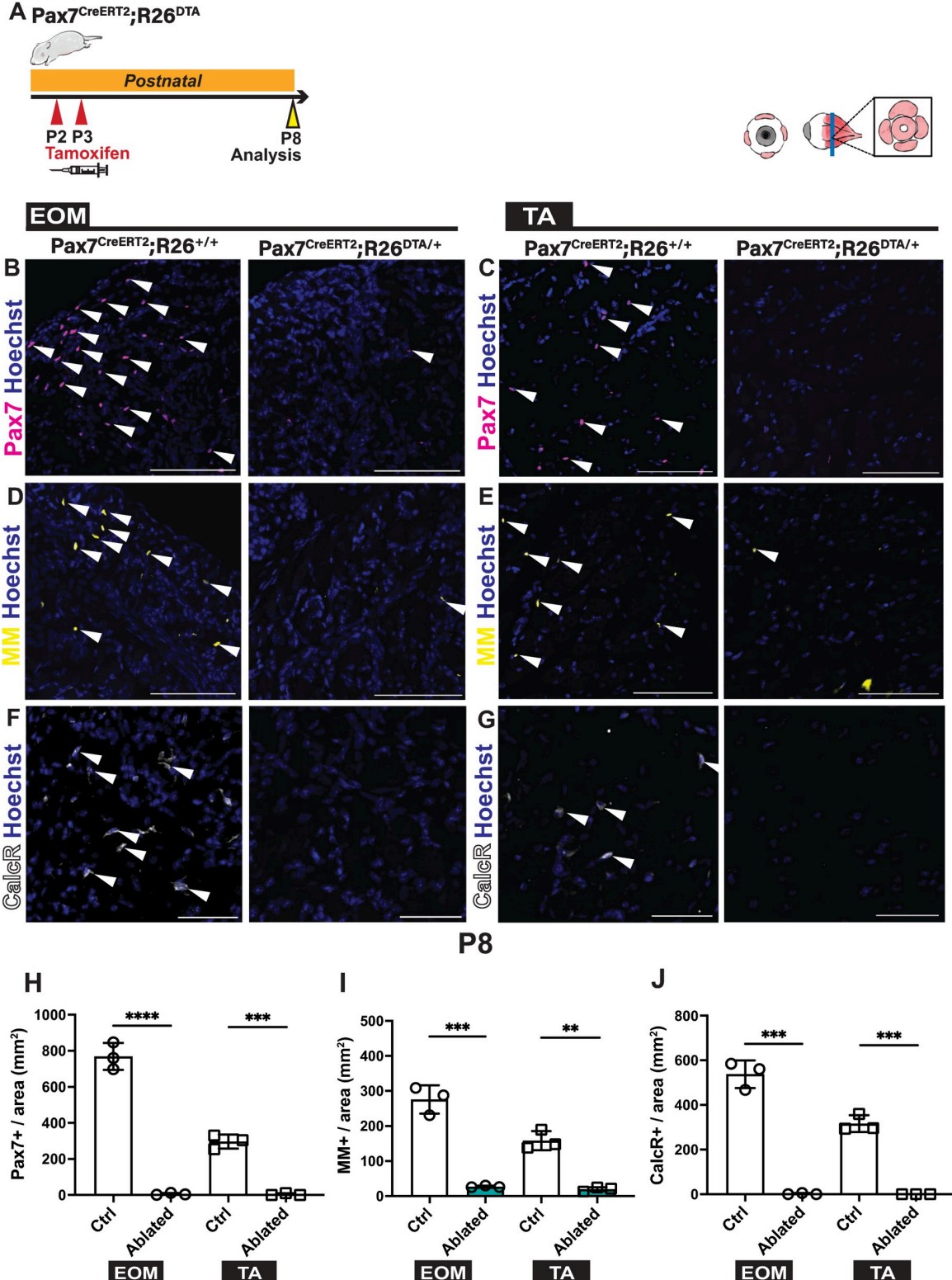

**Fig 5. Postnatal ablation of Pax7+ cells results in major loss of myogenic cells in EOMs.** (A) Scheme indicating timing of Tamoxifen injections for ablation of *Pax7*-expressing cells postnatally. Tamoxifen was injected at P2 and P3, samples were collected at P8. **(B,C)** Immunostaining for Pax7 (magenta) on *Pax7^{CreERT2};R26^{+/+}* (control) and *Pax7^{CreERT2};R26^{DTA/+}* (ablated) EOM and TA sections. White arrowheads indicate Pax7+ cells. **(D,E)** Immunostaining for Myod and Myogenin (MM) (yellow) on *Pax7^{CreERT2};R26^{+/+}* (control) and

*Pax7$^{CreERT2}$;R26$^{DTA/+}$* (ablated) EOMs and TA sections. White arrowheads indicate MM+ cells. **(F,G)** Immunostaining for Calcitonin receptor (CalcR) (white) on *Pax7$^{CreERT2}$;R26$^{+/+}$* (control) and *Pax7$^{CreERT2}$;R26$^{DTA/+}$* (ablated) EOMs and TA sections. White arrowheads indicate CalcR+ cells. **(H)** Number of Pax7+ cells per area from immunostaining in (B, C) (Control, n = 3, Ablated, n = 3). **(I)** Number of MM+ cells per area from immunostaining in (D,E) (Control, n = 3, Ablated, n = 3). **(J)** Number of CalcR+ cells per area from immunostaining in (F,G) (Control, n = 3, Ablated, n = 3). Scale bars: 100μm in (B-E), 50μm in (F,G). Error bars represent mean ± SEM. Two-tailed unpaired Student's t-test. **P<0.01, ***P<0.001. EOM, extraocular muscle; TA, Tibialis anterior. All recti EOMs were assessed.

## Temporal requirement for *Pitx2* in extraocular MuSCs

To assess the role of *Pitx2* specifically in Pax7+ cells, we generated Tamoxifen-inducible *Pax7$^{CreERT2}$; Pitx2$^{flox/flox}$* mice (conditional knockout, cKO). Tamoxifen was induced twice at E11.5 and E12.5, at the onset of *Pax7* expression in developing EOMs (Fig 3B and 3C), and samples were collected at E17.5 (Fig 6A). Immunostaining for Pax7, Pitx2 and MF20 showed that although myofibres were present in the EOMs of *Pax7$^{CreERT2/+}$; Pitx2$^{flox/flox}$* foetuses (Fig 6B), the number of Pax7+ cells was reduced by 31.3% (Fig 6C), and the proliferating population (Ki67+) was reduced by 16.6% (Fig 6D) in the cKO. No significant perturbations were noted in the hindlimb muscles at this stage.

Next, we investigated the role of *Pitx2* in Pax7+ cells postnatally where Tamoxifen was injected at perinatal stages (from P1 to P7, condition 1) or at weaning (from P21 to P25, condition 2) with additional monthly injections until analysis in the adult (4 months) (Fig 6E). The rationale was to assure full *Pitx2* deletion from the time of induction, and to avoid a potential rescue by non-deleted cells. In both conditions, there was no difference in the number of EOM Pax7+ cells between the cKO and controls (Fig 6F and 6G), despite an efficient deletion in the Pax7+ population (S6A–S6D Fig). Of note, a concomitant broad deletion of *Pitx2* in myonuclei was observed when Tamoxifen induction was done at P7 (S6B Fig) but not P20 (S6C Fig). This is consistent with our results (Fig 1C) showing that most of these MuSCs enter quiescence before P20 with myoblast proliferation, fusion and myonuclear accretion taking place prior to that stage. Furthermore, absence of *Pitx2* alone does not result in the loss of Pax7+ MuSCs during homeostasis postnatally. Additionally, we investigated the short-term phenotype following deletion of *Pitx2* in EOM. Here (condition 3), Tamoxifen was injected from P1 to P7, and analysis was done 2 weeks later (S6E Fig). Similarly to conditions 1 and 2 (Fig 6E–6G), there was no significant difference in the number of Pax7+ cells in EOMs between the cKO and controls (S6F Fig). Altogether, our data shows that *Pitx2* is important for emergence of foetal MuSCs in the EOMs, then becomes dispensable at postnatal stages.

## Loss of Pitx2 in MuSCs does not account for sparing of EOM in *mdx* mice

High levels of Pitx2 in MuSCs were proposed to be responsible, at least in part, for sparing of EOMs in Duchenne muscular dystrophy (DMD) [43,46]. Therefore, we generated *mdx$^{βgeo}$; Pax7$^{CreERT2}$; Pitx2$^{flox/flox}$* compound mutant mice (dKO), where *mdx$^{βgeo}$* is a dystrophin null mouse model [59]. Tamoxifen was injected 3 times from P1 to P7, followed by monthly injections (from 2 months) until analysis in the adult (Fig 7A). Immunostaining for Pax7 and Ki67 allowed evaluation of the density and proliferative fraction of Pax7+ MuSCs in EOMs and TA (Fig 7B–7F). The number of Pax7+ cells per area was similarly decreased in the EOMs of *mdx$^{βgeo}$* and dKO mice compared to controls and cKO implying no further perturbation due to removal of Pitx2 function in *mdx$^{βgeo}$* (Fig 7C). Moreover, there were only a few Ki67+Pax7+ cells in EOMs of the models analysed (Fig 7D). In contrast, TA muscle from *mdx$^{βgeo}$* mice and dKO showed a significant increase in the number of Pax7+ cells per area, and percentage of proliferative MuSCs in areas showing fibre damage (Fig 7E and 7F). Similarly, there was a significant increase in Bromodeoxyuridine

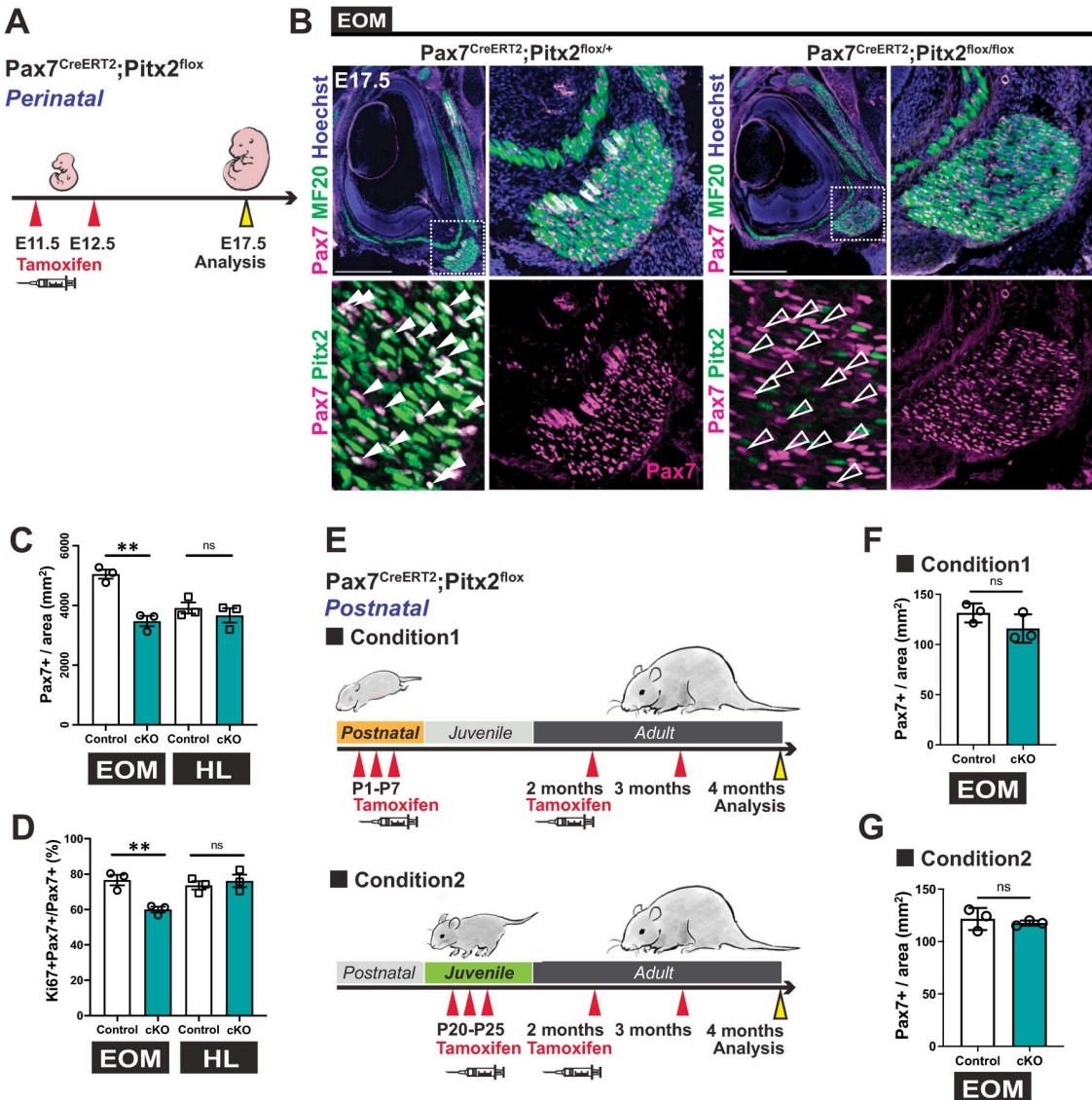

**Fig 6. Timing of inactivation of *Pitx2* differently impairs extraocular MuSCs. (A)** Scheme indicating timing of Tamoxifen injections for invalidation of *Pitx2* during embryogenesis. Tamoxifen was injected at E11.5 and E12.5 to *Pitx2^flox/flox* pregnant females crossed with *Pax7^CreERT2;Pitx2^flox/+* male mice and samples collected at E17.5. **(B)** Immunostaining of E17.5 EOM sections from *Pax7^CreERT2;Pitx2^flox/+* (control) and *Pax7^CreERT2;Pitx2^flox/flox* (cKO) foetuses for Pax7 (magenta), Pitx2 (green) and MF20(cyan). Bottom panels, higher magnification views of the area delimited with dots. White arrowheads indicate Pax7+Pitx2+ cells and white open arrowheads Pax7+Pitx2- cells. **(C)** Number of Pax7+ cells per area in EOM and hindlimb (HL) sections from immunostaining in (B) (n = 3 each). **(D)** Percentage of Ki67+Pax7+ cells over total Pax7+ cells in EOM and HL sections (n = 3 each). **(E)** Scheme indicating timing of Tamoxifen injections for invalidation of Pitx2 with *Pax7^CreERT2*. Condition1: Tamoxifen was administered 3 times from P1-P7, at 2 months, and at 3 months. Condition2: Tamoxifen was administered 3 times from P20-P25, at 2 months, and at 3 months. Samples were collected at 4 months of age. **(F)** Number of Pax7+ cells per area in EOM sections from Condition 1 (n = 3 each). **(G)** Number of Pax7+ cells per area in EOM sections from Condition 2 (n = 3 each). Scale bars: 500μm (B). Error bars represent mean ± SEM. Two-tailed unpaired Student's t-test. ns, non-significant, P>0.05, **P<0.01. EOM, extraocular muscle; TA, Tibialis anterior. All recti EOMs were assessed.

(BrdU) uptake in *mdx^βgeo* TA compared to the EOMs *in vivo* (S7A and S7B Fig). In summary, our data show that MuSCs in EOMs are less proliferative than TA in dystrophic mice, even in the absence of *Pitx2*.

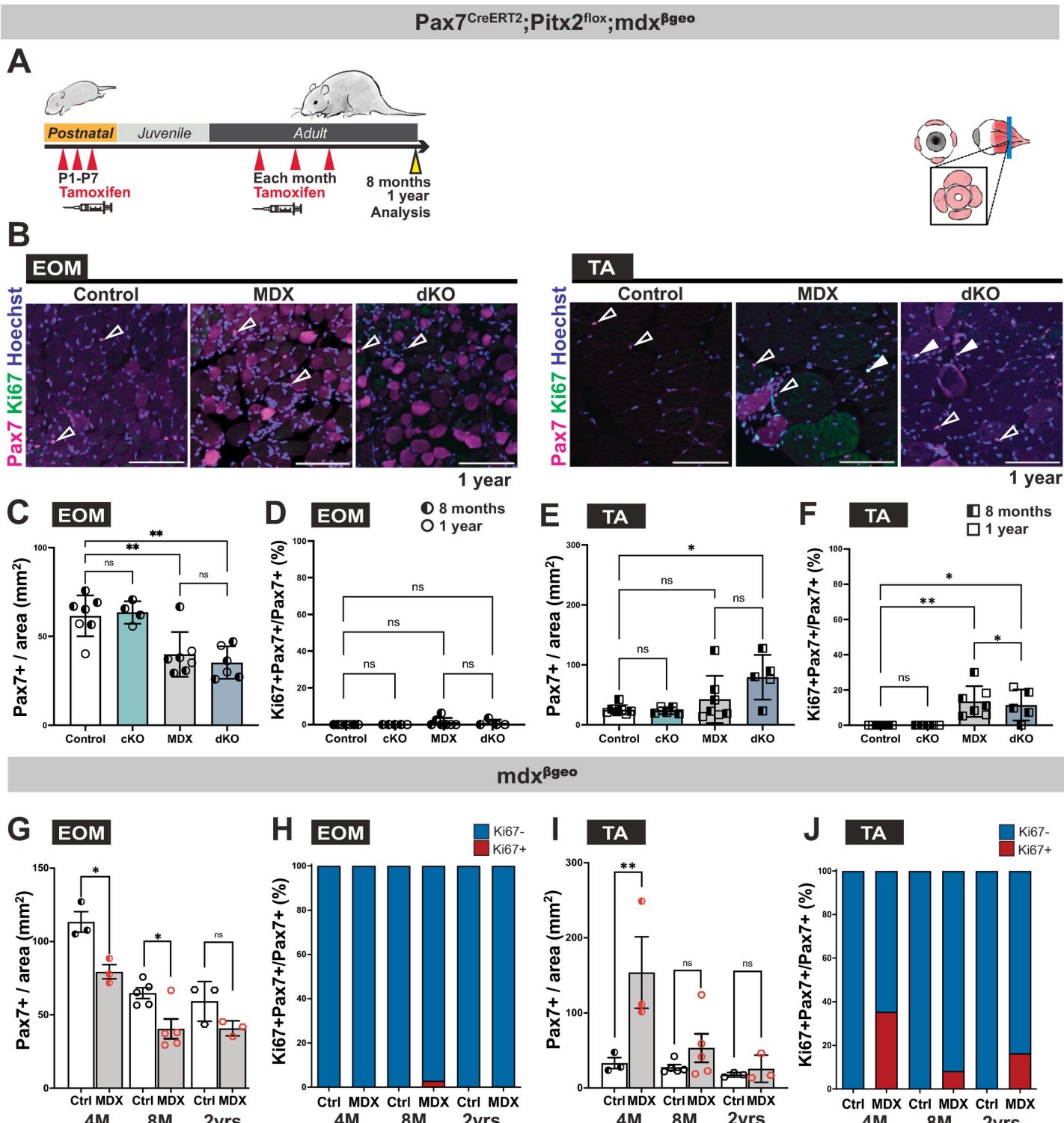

**Fig 7. Absence of proliferation in EOM of *mdx* and *mdx;Pitx2* KO mice. (A)** Scheme indicating timing of Tamoxifen injections on *Pax7^{CreERT2}*;*Pitx2^{flox}*; *mdx^{βgeo}* and control mice at postnatal (3 times from P0-P7) and adult stages (every month). Samples were collected at 8 months and 1 year for analysis. **(B)** Immunostaining of EOM and TA sections from 1 year old control (*Pitx2^{flox/flox}* or *Pitx2^{flox/+}*), cKO (*Pax7^{CreERT2}*;*Pitx2^{flox/flox}*), *mdx^{βgeo}* (MDX) and *Pax7^{CreERT2}*;*Pitx2^{flox/flox}*; *mdx^{βgeo}* (dKO) mice for Pax7 (magenta) and Ki67 (green). White arrowheads point to Pax7+Ki67+ cells; open arrowheads to Pax7+Ki67- cells. **(C)** Number of Pax7+ cells per area on EOM sections from immunostaining in (B) (Control, n = 7, MDX, n = 4; cKO, n = 7; dKO, n = 6). **(D)** Percentage of Ki67+Pax7+ cells over total Pax7+cells on EOM sections from immunostaining in (B) (Control, n = 7, MDX, n = 5; cKO, n = 7; dKO, n = 4). **(E)** Number of Pax7+ cells per area on TA sections from immunostaining in (B)

(Control, n = 7, MDX, n = 5; cKO, n = 7; dKO, n = 5). **(F)** Percentage of Ki67+Pax7+ cells over Pax7+cells on TA sections from immunostaining in (B) (Control, n = 7, MDX, n = 5; cKO, n = 7; dKO, n = 5). In (C),(D),(G),(H), the entire EOMs were counted. In (E),(F),(I),(J), in TA counting was done on a random area for controls and in the damaged/regenerating area of MDX and DKO. 1 year old animals were littermates. 8-month-old mice, dKO and control mice were littermates while MDX were generated in independent litters. All mice were treated with Tamoxifen. **(G)** Number of Pax7+ cells per area on EOM sections from immunostaining at indicated stages from 4 months (4M) to 2 years (2yrs). (4M, n = 3, 8M, n = 5, 2yrs, n = 3). **(H)** Percentage of Ki67+Pax7+ cells over total Pax7+ cells on EOM sections at indicated stages from 4M to 2yrs (4M, n = 3, 8M, n = 5, 2yrs, n = 3). **(I)** Number of Pax7+ cells per area on TA sections at indicated stages from 4M to 2yrs. (4M, n = 3, 8M, n = 5, 2yrs, n = 3). **(J)** Percentage of Ki67+Pax7+ cells over total Pax7+cells on TA sections at indicated stages from 4M to 2yrs. (4M, n = 3, 8M, n = 5, 2yrs, n = 3). Scale bars: 100μm in (B). Error bars represent mean ± SEM. Two-way ANOVA with Dunnett post-hoc test. ns, non-significant, P>0.05, *P<0.05, **P<0.01. EOM, extraocular muscles, TA, Tibialis anterior. All recti EOMs were assessed.

Finally, we assessed the number of centrally nucleated myofibres in the EOMs of $mdx^{\beta geo}$ and dKO mice (S7C and S7D Fig). Only sparse centrally located myonuclei were observed in both conditions. This is at odds with the high numbers of centrally nucleated fibres normally observed in $mdx^{\beta geo}$ TA muscles (S7C Fig), which can reach up to ~60% [60]. Next, we calculated the cross-sectional area of EOM myofibres in the 4 recti muscles from control, $mdx^{\beta geo}$, *Pitx2* cKO and dKO and classified them by size (S7E Fig). While a trend towards larger myofibre size was observed in the EOMs of $mdx^{\beta geo}$ mice, dKO mice showed some heterogeneity with a slight shift towards smaller and larger size fibres (S7E Fig). Altogether, *Pitx2* appears to play a minor role in MuSCs in the EOMs in a dystrophic context in the mouse.

## MuSC number in the EOMs and TA in young and old *mdx* mice

Finally, given the continuous cycles of degeneration and regeneration exhibited by *mdx* mice [61], we performed a proliferation time course analysis in control and $mdx^{\beta geo}$ mice from 4 months to 2 years of age (Fig 7G–7J). During adult stages (4 and 8 months), the number of Pax7+ cells in EOMs in $mdx^{\beta geo}$ was significantly decreased compared to control, whereas an opposite phenotype was observed in the TA (Fig 7G and 7I). In 2 yr old mice, there was no significant difference between control and $mdx^{\beta geo}$, neither in EOM nor TA (Fig 7G and 7I). Notably, throughout all stages examined, Pax7+ cells were not proliferative in the EOMs of $mdx^{\beta geo}$ mice, in contrast to the TA and other cranial muscles such as the masseter (Figs 7H, 7J and S7B). Therefore, the MuSC population exhibited dynamic proliferative properties between young and old dystrophic mice, where those in the EOMs decreased in numbers and remained non-proliferative during ageing.

## Discussion

Pax7 plays a critical role in the maintenance of mouse MuSCs from late foetal stages, and it is the most widely used marker for adult MuSCs and their ancestors from mid-embryogenesis [16,62]. MuSCs in the head express Pax7, yet they also retain the expression of the early mesodermal markers characteristic of each location [10,11,21,23,63]. In this study, we investigated the functional relationships between *Pax7*, *Pitx2* and *Myf5* in the emergence and maintenance of MuSCs. Our data reveal different temporal requirements for *Pitx2* during prenatal and postnatal stages, and during lineage progression in this context (Fig 8).

## Interplay between Pitx2, Myf5 and Pax7 in emergence of extraocular MuSCs

In the trunk, both at the dermomyotome and limb levels, expression of *Pax3* and *Myf5* precede the expression of *Pax7* [1,64–66], with *Pax3* expression being then rapidly downregulated during myogenic cell commitment (i.e. in Myod+ cells). Following the establishment of a muscle anlagen (uncommitted and differentiated cells), *Pax7* is expressed from mid-embryogenesis in the upstream undifferentiated population, and it persists to give rise to adult MuSCs [17,64].

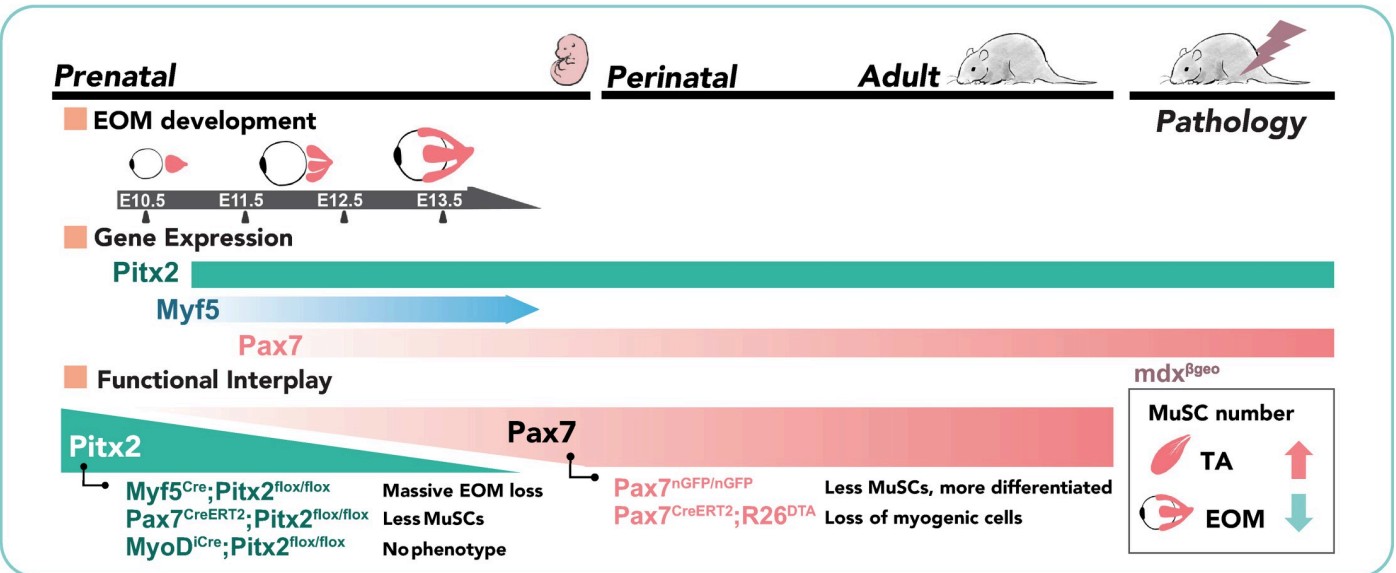

**Fig 8. Stage dependent interplay between Pitx2 and Pax7 in the EOMs.** Schemes summarising the principal findings of this study. *Pitx2* becomes progressively dispensable for regulation of the EOM stem cell pool whereas constitutive inactivation of Pax7 showed a greater loss of muscle stem cells in EOM postnatally compared to limb. Significant loss of myogenic cells occurred in EOMs following DTA-ablation of *Pax7+* cells, with no rescue by a *Pax7*-negative population in this scenario. Finally, significantly less MuSCs are present in EOM compared to the limb in dystrophic *mdx* mice.

In the head of vertebrate models analysed, Pax7+ myogenic cells were reported to emerge de novo, as *Pax3* is not expressed in head MuSCs [21,23,32,33]. Instead, and like the role of *Pax3* in the trunk and limbs, *Pitx2* is critically required cell-autonomously to maintain myogenic progenitors in an immature state, ensure their survival until the end of the embryonic period, and to activate the MRFs [30]. Differently to the role of *Pax3* in the trunk, *Pitx2* expression persists following activation of the MRFs in the developing EOMs [21]. The expression of Pitx2 also in adjacent mesenchymal cells [53,54] has complicated the analysis and interpretation of the cell-autonomous roles of this gene in EOM development. In this context, conditional inactivation of *Pitx2* in neural-crest-derived cells does not affect the specification of the EOM primordia, but it results in severe muscle patterning defects [67].

We demonstrated herein a temporal requirement for *Pitx2* during lineage progression: deletion of this gene in muscle progenitors (*Myf5^Cre*) results in the absence of myofibres and upstream cells in the embryo, while deletion in myoblasts (*Myod^iCre*) does not. Further, deletion of this gene in Pax7+ cells led to a reduction in the number of foetal MuSCs in the EOMs, whereas *Pitx2* function appears to be dispensable for the maintenance of these MuSCs in the adult (Fig 8). In this muscle group, Myf5 and Mrf4 also play critical roles as EOMs and the upstream population are absent in *Myf5;Mrf4* double mutants [21]. Collectively, these observations indicate that EOM specification and differentiation has evolved a unique gene regulatory network where EOM progenitor survival as well as activation of the downstream myogenic factors are dependent on either *Myf5* or *Mrf4*, and that *Pitx2* alone cannot override their role. Therefore, it is possible that the *Pitx2* survival function is progressively relayed onto *Myf5* and/or *Pax7*. A double *Pax7;Pitx2* KO in the MuSC lineage would be required to formally address this hypothesis. Moreover, while Pitx2 binds directly to the promoters of *Myf5* and *Myod* [30], it is unclear whether it can directly activate *Pax7*, or if other TFs are necessary. Of note, sc-RNAseq data from our lab [3] revealed several novel TFs that are specifically expressed in EOM progenitors. Whether those factors are required for the emergence of the myogenic

population in the EOMs requires further investigation. Moreover, a recent study showed that *Six1* and *Six2* genes are required for craniofacial myogenesis by controlling the engagement of unsegmented cranial paraxial mesodermal cells in the myogenic pathway [22]. It remains unclear if expression of these genes in adult MuSCs would be required for *Pax7* maintenance in the EOMs.

## Adult extraocular MuSCs do not proliferate during homeostasis despite coexpression of Pax7 and Pitx2

EOM activated Pax7+ cells have greater proliferative and self-renewal abilities *in vitro* compared to those in the limb [10,68,69]. Moreover, expression of *Pitx2* in postnatal EOM myogenic cells (CD34+/Sca1−/CD31−/CD45− lineage), which is at higher levels than those in the limb, was reported to contribute to the proliferative and stress resistance properties of these cells *in vitro* [9,43]. However, with respect to the proliferative status of extraocular MuSCs *in vivo*, we note several discrepancies. In different species (mouse, rabbit, monkey, and human), MuSCs were suggested to chronically proliferate *in vivo* [9,70–73], but these studies generally lacked co-immunostainings for Pax7 and proliferation markers. Yet, MuSCs appeared to contribute to new myofibre myonuclei in homeostasis at higher frequency in the EOMs than that observed for limb muscles [45,48,70,71,73]. Here, we performed co-immunostaining for the canonical MuSC marker Pax7 and Ki67 in the mouse and found that MuSCs entered quiescence in the EOMs earlier than those in TA. Moreover, and we did not observe proliferative MuSCs in EOMs during adult homeostasis even in BrdU and EdU uptake experiments. Our observations agree with a previous study [69] showing that freshly isolated mouse MuSCs from EOMs are not proliferative. While the origin of the discordance with lineage tracing studies is unclear, one possibility could be that MuSCs in the EOMs contribute to myofibres without cell cycle entry and/or the genetic constructs used as *Cre* drivers [45,48] are unexpectedly expressed in these myofibres.

Despite an earlier entry in quiescence, EOMs have a significantly higher content of Pax7 + MuSCs per area, but not per fibre compared to TA. This agrees with previous data showing higher number of MuSCs per volume unit in the EOMs [46], in line with their smaller fibre size. We also observed a lower fraction of committed and differentiating myoblasts (Myod and Myogenin+) in the EOMs of $Pax7^{nGFP}$ heterozygous controls at E14.5 and P21 compared to the limb. This unique feature could explain, at least in part, the larger MuSC pool in the EOMs. It could result from different dynamics of myogenic lineage progression taking place in EOMs and limb muscles during the growth phase *in vivo* and be related to the deferred differentiation of activated extraocular MuSCs observed *in vitro* [10].

It was also proposed that Pitx2+ myogenic cells do not co-express *Pax7* but express *Myod* in human EOMs, and these cells could then represent a second progenitor population involved in EOM remodeling, repair, and regeneration [43]. Given our observation that more than 92% of Pax7+ MuSCs coexpress *Pitx2*, this might highlight species specific differences or antibody sensitivity and tissue processing issues in the different studies.

## Deletion of Pax7 results in loss of the MuSC population in EOMs

Several studies analysed the limb and trunk phenotypes of constitutive [35–38] and inducible [39–41] *Pax7* mutants. While MuSC progenitors are initially specified in homeostatic conditions [38], the progressive loss of MuSCs in newborn mice appears to be due to proliferation defects of MuSCs and cell death [37,38]. Elimination of *Pax7* in adult MuSCs results in MuSC loss likely due to enhanced differentiation, reduced heterochromatin condensation and severe deficits in muscle regeneration [36,38,40,41]. The phenotypes of *Pax7* null cells during *in vitro*

activation were somewhat more divergent, depending on the system used (isolated myoblasts, myofibres, clonogenic assays) including reduction in the number of myoblasts, increased or reduced differentiation and/or commitment to alternative fates [35–38,40,41].

As EOMs are not amenable to *in vivo* muscle injury, in this study we examined the *in vivo* phenotype of *Pax7^{nGFP/nGFP}* null EOMs compared to heterozygous controls at E14.5 and P21. The perdurance of the GFP allowed us to trace the progenitors and more differentiated cells. We highlight several differences between the EOM and TA muscles (Fig 8). First, a greater reduction of GFP+ cells per area was noted in *Pax7* null EOMs compared to TA, suggestive of a more relevant role for *Pax7* in the maintenance of MuSCs without a history of *Pax3* expression. Second, no differences in the number of GFP+ cells per area nor in the proportion of Pitx2+GFP+ cells were observed in the foetal period between control and mutant EOMs, suggesting that factors other than Pitx2 (and Pax3) may ensure the specification of this cell population during embryonic development. In this context, a higher number of dying cells per muscle area were observed in *Pax7* null EOMs compared to controls at P1, suggesting that this might be a critical period where compensation by other factors is minimal. Finally, previous studies showed that Pax7 is required for limb myogenic cells to express genes normally associated with functional MuSCs such as Syndecan4 and CD34 [36,38]. Here, we further show that expression of CalcR, another MuSC marker [50], was absent in EOMs and TA lacking Pax7. In addition, while the fraction of committed GFP+ cells in *Pax7* null EOMs display a marked increase compared to controls, no significant changes were observed in mutant and control TA at P21. This observation probably reflects differences on the pace of lineage progression between these muscles *in vivo*. It would be of interest in future studies to assess whether the downstream targets of *Pax7* in MuSCs of cranial and somite origin are similar and to identify complementary transcription factors that confer these distinct MuSC phenotypes.

## Absence of proliferation in EOMs in *mdx* and *mdx;Pitx2* KO mice

Dystrophin protein is expressed in differentiated myofibres where it is required for sarcolemmal integrity, but it was also reported to be expressed in activated MuSCs, where it regulates MuSC polarity and the mode of cell divisions [74,75]. Moreover, several mechanisms of MuSC dysfunction have been described in DMD [76], including severe proliferation defects and premature senescence of chronically activated MuSCs [42,77–79]. Notably, in a preclinical rat model of DMD, expression of thyroid-stimulating hormone receptor (Tshr) was shown to protect MuSCs in the EOMs from entering senescence [42].

In this study, we found that the number of Pax7+ cells/area were reduced in EOMs of *mdx* and *mdx;Pitx2* KO mice, and that these cells were not proliferative (Fig 8). The *mdx* mice have a relatively normal lifespan and have a milder phenotype compared to rats, dogs, and human [80,81]. Thus, the absence of proliferation of *mdx* MuSCs in the EOMs might reflect a complete sparing of these muscles in the murine model. Alternatively, it might reflect activation and fusion of the MuSC population without prior entry in the cell cycle given that the number of Pax7+ cells/area was similarly reduced in *mdx* and *mdx;Pitx2* dKO mice compared to control EOMs. Moreover, as MuSCs in *Pax7^{CreERT2};Pitx2^{fl/fl}* EOMs were unaffected when *Pitx2* was deleted postnatally, these data suggest that the reduction in the number of MuSCs in dKO EOMs is *mdx* but not *Pitx2* dependent. A recent study showed that depletion of MuSCs attenuates pathology in muscular dystrophy [82]. Mechanistically, it was proposed that reactivation of the foetal program via Myod might provoke destabilisation of myofibres in *mdx* mice. Following this logic, one possibility is that less MuSC proliferation in EOMs limits destabilisation of *mdx* myofibres by minimising the input of Myod+ myoblasts into the myofibre. Further studies should clarify the relevance of this mechanism in the EOMs.

Finally, some insights into the potential role of *Pitx2* in the sparing of EOMs in DMD come from a study showing that in the absence of *Pitx2* expression in myofibres of *mdx4cv* mice, the EOMs succumb to dystrophic changes that are even more severe than those seen in the limb muscles of the same mice [46]. In the present study, we deleted *Pitx2* in MuSCs of $mdx^{\beta geo}$ mice at an early postnatal stage (<P7), where considerable fusion is ongoing and thus MuSC-derived myoblasts actively contribute to new myonuclei. While this resulted in a small percentage of centrally nucleated fibres and a more heterogeneous distribution of fibre sizes (shift towards smaller and larger fibres), the resulting phenotype was less severe than when *Pitx2* was deleted with a constitutive myofibre *Cre* driver [46]. However, the latter study was performed with 18 months old mice, therefore a long-term phenotype upon deletion of *Pitx2* in MuSCs cannot be excluded. Whether these differences could also come from the *mdx* models used is unclear. Performing deletion of *Pitx2* prenatally in MuSCs or timed deletions within myofibres using a conditional myofibre *Cre* driver mouse would help elucidate the relative and roles of *Pitx2* in each compartment in the dystrophic condition.

In summary, how some muscles are spared in myopathies, such as the EOMs in dystrophic mice, remains unresolved and under debate. Our studies on the prenatal and postnatal development of susceptible and spared muscles provides insights into this process regarding the roles of key transcription factors that specify and maintain the MuSC populations in these muscles.

## Materials and methods

### Ethics statement

Animals were handled according to national and European Community guidelines and an ethics committee of the Institut Pasteur (CETEA, Comité d'Ethique en Expérimentation Animale) approved protocols (APAFIS#41051–202302204207082).

### Mouse strains

The following strains were previously described: $Myf5^{Cre}$ [55], $Myf5^{nlacZ}$ [52], $Pax7^{CreERT2}$ [57], $Myod^{iCre}$ [56], $mdx^{\beta geo}$ [59], $Pax7^{nGFP}$ [21], $R26^{DTA}$ [58], $R26^{tdTomato}$ [83], $R26^{mTmG}$ [84] and $Pitx2^{flox}$ [27], in which the DNA binding homedomain region common to Pitx2a/b/c isoforms is flanked with LoxP sites. Mice were kept on a mixed genetic background (B6D2F1, Janvier Labs). Mouse embryos and foetuses were collected at embryonic day (E) E14.5 and E17.5, with noon on the day of the vaginal plug considered as E0.5. To generate conditional KOs, $Pax7^{CreERT2/CreERT2}$; $Pitx2^{flox/+}$, $Myf5^{Cre/+}$; $Pitx2^{flox/+}$ and $Myod^{iCre/+}$; $Pitx2^{flox/+}$ males were crossed with $Pitx2^{flox/flox}$ or $Pitx2^{flox/+}$ females. Of note, $Myf5^{Cre/+}$;$Pitx2^{flox/+}$ males crossed to $Pitx2^{flox/+}$ females resulted in litters with $Pitx2^{flox/null}$ phenotypes, suggesting that one of the flox alleles underwent recombination in the germline. To generate dKO ($Pax7^{CreERT2/+}$; $Pitx2^{flox/+}$; $mdx^{\beta geo}$), $Pax7^{CreERT2/CreERT2}$; $Pitx2^{flox/+}$ males were crossed with $Pitx2^{flox/flox}$; $mdx^{\beta geo/+}$ or $Pitx2^{flox/+}$; $mdx^{\beta geo/+}$ females. A list of all genetic models is provided in **S1 Table**.

### Immunofluorescence

Embryonic tissues were fixed for 2h in 4% paraformaldehyde (PFA) in PBS (Electron Microscopy Sciences, Cat #:15710) with 0.5% Triton X-100 at 4°C. Adult muscles were fixed for 2h in 1% PFA in PBS with 0.1% Triton X-100 at 4°C. After PBS washes for a few hours, samples were equilibrated in 30% sucrose in PBS overnight at 4°C then embedded in OCT compound (Sakura Finetek, Cat #:4583). Cryosections (16μm) were permeabilized with 0.5% Triton in PBS for 5 min at RT and blocked for 1h at RT in Blocking solution (10% Goat serum, 1% BSA

and 0.5% Triton X-100 in PBS). Primary antibodies were added in blocking buffer overnight at 4°C. Following washes in PBS-T (0.05% Tween20 in PBS), secondary antibodies and Hoechst (Thermo Scientific, Cat. #:H3570, dilution 1:10000) were diluted in Blocking solution and incubated for 1h at RT. Primary and secondary antibodies used are listed in **S2 Table**. Images were acquired by Spinning disk Ti2E (Nikon). WMIF and clearing were performed as described [53]. Images were acquired with a confocal microscope (LSM700) and 3D reconstructions performed on Imaris.

## Tamoxifen administration

Tamoxifen injections were done to induce $Cre^{ERT2}$-mediated recombination in $Pax7^{CreERT2}$; $Pitx2^{flox/flox}$ mice and $Pax7^{CreERT2};R26^{DTA}$ mice. A solution of 15-20mg/ml Tamoxifen (Sigma-Aldrich, Cat# T5648) in 5% ethanol in sunflower oil was prepared by vortexing and rolling at 4°C in the dark and kept for up to one week at 4°C. For the induction at embryonic stages, pregnant female mice were treated with 150µl tamoxifen solution (20 mg/ml) by gavage at the indicated timepoints. For induction at P1, pups were injected subcutaneously with 10µl tamoxifen solution (15mg/ml) three times between P1 and P7. For induction at P20, pups were injected intraperitoneally with 50µl tamoxifen solution (20mg/ml) three times between P20 and P25. Adult mice were treated with 150µl tamoxifen solution (20mg/ml) by gavage monthly from 2 months until the time of analysis. Tamoxifen was injected in all animals regardless of their genotypes ($Pitx2^{flox/+}$, $Pitx2^{flox/flox}$, $mdx^{\beta geo}$, cKO and dKO). We made great efforts to generate $Pax7^{CreERT2};$ $Pitx2^{flox/flox}$ mice induced at E11.5 and E12.5 to be analysed postnatally. Tamoxifen administration during embryogenesis can cause abortions and mortality of pups at birth due to difficult delivery [85,86]. To ameliorate negative effects of tamoxifen, pregnant females were injected with 60 µl of a mixture of 20 mg/ml Tamoxifen and 16.6 mg/ml Progesterone (Sigma Aldrich). Even under those conditions, no living pups were obtained.

## Assessment of cell proliferation by EdU and BrdU uptake

For EdU uptake experiments 0.5mg/ml of EdU (E10187) and 10mg/ml sucrose were added in drinking water for 2 weeks. The solution was changed every 3 days. Tissues were collected after 2 weeks and processed as above. EdU was detected after staining using the Click-iT Plus EdU Cell Proliferation Kit according to manufacturer instructions (ThermoFisher, C10640). For BrdU uptake experiments, mice were first injected with 5µg/g body weight of BrdU (in saline; Sigma B5002) intraperitoneally, followed by 7 days administration in drinking water at 0.5mg/ml with 2mg/ml sucrose. MuSCs were isolated by FACS, plated and BrdU detected by immunostaining prior unmasking with 2N HCl 20 min at room temperature and neutralized with 0.1M sodium tetraborate.

## Image analysis and statistics

Quantifications were performed using Fiji (https://imagej.net/software/fiji/) and Qupath (https://qupath.github.io/). Barplots were generated using Prism (https://www.graphpad.com/features). Fibre sizes and numbers were measured automatically by Cellpose (https://cellpose.readthedocs.io/en/latest/) and Fiji. All data are presented as the mean ± standard error of the mean (SEM). Statistical significance was assessed using Student's t-test and two-way ANOVA with post-hoc Dunnett test depending on the type of comparisons. All recti EOMs were scored in each experiment and no differences observed among them.

## Supporting information

**S1 Fig. Postnatal analysis of Pax7+ MuSCs in the EOMs. (A)** Immunostaining of P7, P14, P20 and 4 months old EOM sections for (A) Pax7 (white) and Hoechst nuclei staining or (B) Laminin (white). Global layer (gl) and orbital layer (ol). Higher magnification views of the areas delimited with dots. **(B)** Scheme of the experiment. EdU was administered in drinking water to adult mice for 2 weeks. **(C)** Immunostaining for Pax7 and EdU detection at the EOM, TA, and DIA level as per the experiment in B. White arrowheads indicate Pax7+EdU+ cells. White open arrowheads indicate Pax7+EdU- cells. **(D)** Percentage of EdU+Pax7+ cells over total Pax7+cells on EOM and TA sections (n = 4 each). Scale bars: 200μm in (A), 100μm in (C). Error bars represent mean ± SEM. Two-tailed unpaired Student's t-test. **P<0.01. EOM, extraocular muscle; TA, Tibialis anterior; DIA, diaphragm. All recti EOMs were assessed. (JPG)

**S2 Fig. Role of *Pax7* in the maintenance of extraocular myogenic cells. (A, B)** Immunostaining for GFP (green) on EOM sections from $Pax7^{nGFP/+}$ and $Pax7^{nGFP/nGFP}$ mice at P20. **(A', B')** Higher magnification views of the area delimited with dots. White arrowheads indicate GFP+ cells in global layer (gl); open arrowheads indicate GFP+ cells in orbital layer (ol). **(C)** Number of GFP+ cells per 100 fibres at P20 (n = 5 each). **(D,E)** Immunostaining for GFP (green) on EOM and HL sections from $Pax7^{nGFP/+}$ and $Pax7^{nGFP/nGFP}$ mice at E14.5. **(F)** Number of GFP+ cells per area from immunostaining in (D,E) (EOM n = 4, TA n = 3). **(G,H)** Immunostaining for GFP (green) together with Myod and Myogenin (MM) (magenta) on EOM and HL sections from $Pax7^{nGFP/+}$ and $Pax7^{nGFP/nGFP}$ mice at E14.5. White arrowheads indicate MM+GFP+ cells. **(I)** Percentage of MM+GFP+ cells over total GFP+ population from immunostaining in (G,H) (n = 3 each). **(J, K)** Immunostaining for GFP (green) and Pitx2 (magenta) on EOM and HL sections from $Pax7^{nGFP/+}$ and $Pax7^{nGFP/nGFP}$ mice at E14.5. White arrowheads indicate Pitx2+GFP+ cells. **(L)** Percentage of Pitx2+GFP+ cells over total GFP+ population from immunostaining in (J,K) (n = 3 each). **(M)** Percentage of GFP-Pitx2+ cells over the total number of Pitx2+ interstitial cells on EOM sections at P20 (n = 3 each). Laminin was used for counting cells in the interstitium. **(N-Q)** Immunostaining for GFP (green), cleaved-caspase3 (Red) and MF20 (cyan) at the level of the EOMs from $Pax7^{nGFP/+}$ and $Pax7^{nGFP/nGFP}$ mice at E14.5 (N,O) and P1 (P,Q). Higher magnification views of the area delimited with dots. White arrowheads indicate Cleaved-Caspase3+ cells. **(R, S)** Immunostaining for GFP (green), Cleaved-Caspase3 (Red) and dystrophin (white) on EOM sections from $Pax7^{nGFP/+}$ and $Pax7^{nGFP/nGFP}$ mice at P20. Higher magnification views of the area delimited with dots. White arrowheads indicate Cleaved-Caspase3+ cells. **(T)** Number of Cleaved-Caspase3+ cells per area from immunostaining in (N-S) (n = 3 per stage). G/+: $Pax7^{nGFP/+}$, G/G: $Pax7^{nGFP/nGFP}$. Scale bars: 1000μm in (A, B), 200μm in (D-K,N-Q), 100μm in (R,S). Error bars represent mean ± SEM. Two-tailed unpaired Student's t-test. ns, non-significant, P>0.05, *P<0.05, ***P<0.001, ****P<0.0001. EOM, extraocular muscle; HL, hindlimb; TA, Tibialis anterior. All recti EOMs were assessed. (JPG)

**S3 Fig. Loss of EOMs in $Myf5^{Cre}$; $Pitx2^{flox/flox}$ mice. (A)** Single Z-section of the respective WMIF segmented volumes of whole-mount immunostaining of $Myf5^{nlacZ}$ EOM anlage for Pitx2 (magenta) and β-gal (green) at E11.5, E12.0 from Fig 3B. Right panels, higher magnification views of the area delimited with white line or dots. **(B,C)** Immunostaining of EOM sections from $Myf5^{Cre};Pitx2^{flox/+}$ (control) and $Myf5^{Cre};Pitx2^{flox/flox}$ (KO) at E17.5 for Pax7 (magenta), Pitx2 (green) and MF20 (cyan). White arrowheads indicate Pax7+Pitx2+ cells; open arrowheads indicate Pax7+Pitx2- cells. **(D)** Immunostaining of EOM sections from

$Myf5^{Cre}$;$Pitx2^{flox/+}$ (control) and $Myf5^{Cre}$;$Pitx2^{flox/flox}$ (KO) at E17.5 for Myod and Myogenin (magenta), Pitx2 (green) and MF20 (cyan). Higher magnification views of the area delimited with dots. Samples were evaluated in ventral (B) and dorsal (C,D) anatomical locations. Scale bars: 500μm (B-D). EOM, extraocular muscle. All recti EOMs were assessed.
(JPG)

**S4 Fig. Recombination frequency of cells from $Myod^{iCre}$;$R26^{Tomato}$ lineage tracing in EOMs.** (**A**) Immunostaining of EOM sections from $Myod^{iCre}$;$R26^{Tomato}$ at E14.5 for Pax7 (green), Tomato (Tom, red), with Hoechst (blue). White arrowheads indicate Tom+Pax7+ cells; open arrowheads indicate Tom-Pax7+ cells. (**B**) Percentage of Tom+ and negative Pax7+ cells in EOM from immunostaining in (A) (n = 3 each). Scale bar: 100μm (A). Error bars represent mean ± SEM. Two-tailed unpaired Student's t-test. ****P<0.0001.
(JPG)

**S5 Fig. Ablation of Pax7-expressing cells results in major loss of Pitx2+ cells in the EOMs.** (**A,B**) Immunostaining for Pitx2 (green) on $Pax7^{CreERT2}$;$R26^{+/+}$ (control) and $Pax7^{CreERT2}$; $R26^{DTA/+}$ (ablated) EOM and TA sections at P8. White arrow heads indicate Pitx2+ cells. (**C, D**) Immunostaining for Laminin (white) on $Pax7^{CreERT2}$;$R26^{+/+}$ (control) and $Pax7^{CreERT2}$; $R26^{DTA/+}$ (ablated) EOM and TA sections at P8. Scale bars: 100μm in (A, B), 50μm in (C,D). EOM, extraocular muscle, TA, Tibialis anterior. All recti EOMs were assessed.
(JPG)

**S6 Fig. Evaluation of *Pitx2* deletion efficiency with $Pax7^{CreERT2}$ mice. (A-C)** Immunostaining of EOM sections from 4 months old Control ($Pitx2^{flox/+}$) and $Pax7^{CreERT2}$;$Pitx2^{flox/flox}$ (cKO) mice for Pax7 (magenta) and Pitx2 (green). Condition 1 (❶, induction from P1-P7, then monthly injections, samples collected at 4 months) and Condition 2 (❷, induction from P20-P25, then monthly injections, samples collected at 4 months). Higher magnification views as insets in (A', B', C'). White arrowheads indicate Pax7+Pitx2+ cells; open arrowheads indicate Pax7+Pitx2- cells, yellow arrowheads indicate Pitx2+ myonuclei. (**D**) Percentage of Pitx2 +Pax7+ cells over total Pax7+ cells in EOMs from induction Condition 1 (❶) and Condition 2 (❷) (WT, cKO n = 3 each). (**E**) Scheme indicating timing of Tamoxifen injections for invalidation of *Pitx2* with $Pax7^{CreERT2}$. Condition3 (Tamoxifen induction 3 times from P1-P7, samples collected at P20). (**F**) Number of Pax7+ cells per area in EOM sections from Condition 3 (n = 3 each). Scale bars: 100μm (A-C). Error bars represent mean ± SEM. Two-tailed unpaired Student's t-test. ns, non-significant, P>0.05, ***P<0.001, ****P<0.0001. EOM, extraocular muscle. All recti EOMs were assessed.
(JPG)

**S7 Fig. MuSC and myofibre phenotypes in *mdx* mice. (A)** Experimental scheme for BrdU uptake. A pulse of BrdU was administered intraperitoneally to adult mice followed by 7 days in drinking water. (**B**) Percentage of BrdU+Pax7+ cells over total Pax7+ population isolated from EOMs, TA, MASS. (**C**) Immunostaining of EOM and TA sections from Control ($Pitx2^{flox/flox}$ or $Pitx2^{flox/+}$), Pax7$^{CreERT2}$;Pitx2$^{flox/flox}$ (cKO), MDX and MDX Pitx2cKO (dKO) mice at 8 months of age for laminin (white) and Hoechst nuclei staining (blue). Right panels (i, ii), higher magnification views of the area delimited with dots. (i) Sporadic EOM regions containing central myonuclei, (ii) normal EOM regions. Yellow arrowheads indicate central myonuclei. (**D**) Number of centrally nucleated myofibres in EOMs from immunostaining in (C). (**E**) Distribution of extraocular myofibre size from immunostaining in (C) (Control, cKO and MDX, n = 3; dKO, n = 4). Red arrowheads indicate fibres with a cross-sectional area (CSA) over 2000 μm. Scale bars: (C) 1000μm in lower magnification, 100μm in higher magnification views. Error bars represent mean ± SEM. (B) Two-tailed unpaired Student's t-test. (E) Two-way ANOVA with

Dunnett post-hoc test. ns, non-significant, *P<0.05, ***P<0.001. EOM, extraocular muscle, TA, tibialis anterior, MASS, masseter. All recti EOMs were assessed.
(JPG)

**S1 Table. Genetic models used in this study.**
(XLSX)

**S2 Table. Key resource table.**
(XLSX)

## Acknowledgments

We thank Francesca Pala, Dounia Bouragba and Sayna Miri for initial contributions to this study as well as Melania Murolo and Pierre-Henri Commere for help on data analysis. We gratefully acknowledge the UtechS Photonic BioImaging (Imagopole), C2RT, Institut Pasteur, for support in conducting this study. This platform is independently supported by the French National Research Agency (France BioImaging; ANR-10–INSB–04; Investments for the Future).

## Author Contributions

**Conceptualization:** Glenda Comai, Shahragim Tajbakhsh.

**Data curation:** Glenda Comai.

**Formal analysis:** Mao Kuriki, Amaury Korb, Glenda Comai.

**Funding acquisition:** Glenda Comai, Shahragim Tajbakhsh.

**Investigation:** Mao Kuriki, Amaury Korb, Glenda Comai.

**Methodology:** Mao Kuriki.

**Project administration:** Glenda Comai, Shahragim Tajbakhsh.

**Resources:** Shahragim Tajbakhsh.

**Supervision:** Glenda Comai, Shahragim Tajbakhsh.

**Visualization:** Mao Kuriki.

**Writing – original draft:** Mao Kuriki, Glenda Comai.

**Writing – review & editing:** Mao Kuriki, Glenda Comai, Shahragim Tajbakhsh.

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
