## [Decision Letter · Decision Letter 0]

24 Oct 2023

Dear Dr. Comai,

Thank you for your recent submission to PLOS Genetics entitled "Interplay between Pitx2 and Pax7 temporally governs specification of extraocular muscle progenitors” (PGENETICS-D-23-00956).

The manuscript was fully evaluated at the editorial level and by independent peer reviewers. The reviewers appreciated the attention to an important problem, but raised some concerns about the current manuscript. Based on the reviews, we will not be able to accept this version of the manuscript, but we would be willing to review a revised version. We cannot, of course, promise publication at that time.

If you decide to revise the manuscript for further consideration at PLOS Genetics, please aim to resubmit within the next 60 days, unless it will take extra time to address the concerns of the reviewers, in which case we would appreciate an expected resubmission date by email to plosgenetics@plos.org.

Thank you for submitting your manuscript to PLOS Genetics and we look forward to receiving your revised manuscript.

Yours sincerely,

Peter S Zammit

Guest Editor

PLOS Genetics

Gregory Barsh

Editor-in-Chief

PLOS Genetics

Reviewer's Responses to Questions

**Comments to the Authors:**

Reviewer #1: This manuscript uses a large number of genetic mutant mouse lines to understand muscle progenitor cells in the extraocular muscles compared to tibialis anterior muscles, from embryonic d14.5 to 4 months postnatal age. I have a number of concerns that relate specifically to the way in which cells were defined, counted, and reported.

1. It is always a bit worrisome when results differ from a large number of published studies from multiple laboratories. There are a huge number of genetic mice generated. There is no discussion of compensatory changes from these various manipulations. This needs to be considered. There are other mononuclear cells with myogenic potential that have been described in skeletal muscle. These also need to be considered and discussed.

2. The Results contain a large amount of introductory material, which should be in the Introduction, and a lot of interpretation and discussion that should be in the Discussion. These should be moved to the appropriate locations.

3. As there are so many genetically manipulated mice, a table describing each of the mice genotypes used would help the reader.

4. In Figure 1E, you state that 90% of the Pax7+ cells express Pitx2. Assuming that green is Pitx2 and pink is Pax7 and co-expressing cells are white, I count 52 nuclei that are Pitx2 positive at P20, 13 nuclei that are Pax7 positive at P20, and 3 white nuclei. This would mean there were about 5% co-expressing cells, 20% were Pax7 positive, and 80% were Pitx2-positive. Similar there are even fewer Pax7 positive cells at P14 and P7. By reporting these numbers as the percent of Pax7 cells that were also Pax7+ and Pitx2+ you don’t accurately represent the cell populations in your tissue samples. Similarly in Figure 1C you show the percent of Pax7 positive cells that are Ki67 positive and Pax7 positive. This is not an accurate description either. If Pax7 is a marker of quiescent satellite cells, then one would expect them to be quiescent (according to Seale et al. 2000 and many other papers).

5. I am very confused about the data in figure 2. If there are no Pax7 positive cells, what are the GFP-expressing cells? (although despite your counts, they sure appear to be more numerous in the sections of EOM fibers). On page 6, about half-way down the page (line numbers would have been helpful), you say that expression of MyoD+/Myogenin+ in cells that are GFP+ indicate “depletion of the stem cell population”. First, this is an interpretation, not a fact. I just moving these sorts of conjectures to the Discussion. Two, it is unclear why these cells are expressing these myogenic markers of differentiation. What pathway could you invoke to suggest this? Second, you don’t really mean “stem cells”, as these are not multipotent. The whole experiment described in Figure 2 is really confusing.

6. Calcitonin receptor is expressed on cells involved in angiogenesis and on immune cells. It is insufficient by itself to give an identity to cells to these “formerly Pax7+ cells”.

7. Similarly you describe the population of MyoD+/Myogenin+ that are green – about ¾ of the page down - to indicate a more rapid transition from stem to committed and differentiated cells. Again your interpretation, not a fact. Move these types of statements to the Discussion.

8. In Figure 2J, K, and L, again you are assessing the percent of green cells that are green and pink. From the images, however, it is clear that as a percent of pink cells, there aren’t very many green ones. Please represent the data to more accurately depict the actual data. There are actually very few pitx2 positive cells that also express the GFP, and in TA there are almost no green cells whatsoever.

9. In figure 3, you indicate that “virtually all” of the cells express both Myf5 and Pitx2. Please show an overlay of the Pitx2 and beta-gal staining at sufficient power where cells can be seen.

10. In Figure 3D and E, I don’t see that all the Pax7+ cells were also beta-gal positive. I see mostly green cells, some pink cells, but I don’t see them co-expressing (as I assume they would then be white.) Is this image not typical?

11. Page 9, paragraph 3 line 6: change “depletion” to “decrease”, since there appear to be pitx2 positive cells still present.

12. In the Discussion, page 12, paragraph 2, please add in the appropriate other references that loss of Pitx2 results in loss of extraocular muscles (Zacharias et al., Diehl et al.) and add the references further down that same page that show that MyoD is needed for progression along the myogenic differentiation lineage (e.g. Zacharias et al 2011; L’Honore et al. 2010), etc.

13. The Methods state that significance was determined by Students’ t-tests. You cannot use Students’ t-tests on data with more than 2 comparisons e.g. Figure 1C, 1D, Figure 2 C, F, I, L, Figure 3E, Figure 5C, D, F, G, Figure6, C, D, E, F, and in the bar graphs of the supplemental figures. These need to be redone using ANOVA and posthoc t-tests if significance is shown.

Reviewer #2: This is a well written manuscript with great presented research. Most of my comments are minor and many of them are stylistic.

1. Consider adding to the introduction a few sentences about the EOMs. It is mentioned in Fig 1 that there are 4 recti and 2 obliquus but it would be helpful to mention this also in the introduction.

2. I was wondering if there were any differences in the results between the EOMs as they derive from multiple myogenic sources. I haven't seen any differences in the pictures. Perhaps a sentence or two could be added in the results to address this shortly, even if it is just 'There were no differences".

3. A few abbreviations were not introduced before 1st time use.

4. Some abbreviations should be updated in context of the text. These are suggestions:

4A. EOM MuSCs change to EO MuSCs (introduce EO) as M and Mu are both muscle

4B. EOM myofibers change to EO myofibers as muscle myofibers is redundant in my opinion

4C. EOM and TA muscles change to EOMs and TA or to EO and TA muscles

5. Use either fetus or foetus but not both in the manuscript.

6. Consider adding abbreviation explanations to the Figure legends so that they can stand alone (independent of the manuscript).

7. Consider changing EOM to EOMs in all figures as most of them show more than 1 EOM.

8. Throughout the text are several occasions where EOMs would fit better than EOM. I added the s where I thought it would be appropriate but leave it to the authors to decide if they want to keep it or not.

9. other very minor things were changed using track changes in the attached PDF

Reviewer #3: This manuscript by Kuriki et al. described the stage specific role of Pitx2 and Pax7 in the specification of EOM progenitors during muscle development and postnatal muscle growth, based on a series of immunohistological assays. Although EOM is well known as a sparing muscle for pathology of muscular dystrophy, the underlying mechanism is largely unknown. Using Cre-driver mouse lines, the authors show that Pitx2 is required to generate (or maintain) myogenic progenitors in the developmental stage but not in the postnatal and juvenile stages. The number of muscle stem cells in EOM is more significantly reduced by Pax7 inactivation compared to those of TA muscle. EOM progenitors tend to enter a quiescent state earlier than those in TA muscles in the postnatal stages. Pax7+ cells are less proliferative in EOM of mdx mice compared to those of TA muscle, which is probably in a Pitx2-independent manner. Overall, this study further characterizes the heterogeneity of muscle stem cells among muscles, providing new insights into pathogenesis of muscular dystrophy. Although the findings are very interesting and the manuscript is well-written, there are some points that need to be mentioned or discussed before publication.

Figure2: GFP-expressing cells in EOM, but not TA, tend to commit to myogenic differentiation when Pax7 is removed. Is increased apoptosis involved in Pax7-KO EOM cells at P20?

Figure 2K: an arrowhead is mislocated.

Figure 5: in Figure 5A-D, did the authors check whether the decrease in the Pax7+ cell numbers persists in the postnatal muscle growth and adult stages (e.g., in 2-month-old mice)? In Figure 5E-G, how about Pax7+ cell numbers at the earlier time points after TAM treatment (e.g., at day 5 after TAM)? Please discuss these points.

Figure 6: the Pax7+cell population of the mdx mice was evaluated with mixed ages ranging from 4 to 12 months of age. I guess that the authors used as many mdx mice as possible for this study. Are there any aging effects on EOM Pax7+cells of mdx mice? If so the authors should carefully interpret these results. Please discuss this point.

Figure 6: EOM is well known to be spared in Duchenne muscular dystrophy. The authors found that muscle stem cells are less proliferative in EOM of mdx compared to those of limb muscle. Is this phenomenon related to a recent study revealing that depletion of muscle stem cells attenuates pathology in muscular dystrophy (Nat Commun 2022 May 26;13(1):2940)?

Mouse strains in Methods: "mdx-βgeo (Wertz and Füchtbauer, 1998)" is duplicated.

Option: a schematic summary of the findings might be helpful for readers’ understanding.

**Have all data underlying the figures and results presented in the manuscript been provided?**

Reviewer #1: **No: **They state that they have not shared any of their data.

Reviewer #2: Yes

Reviewer #3: Yes

PLOS authors have the option to publish the peer review history of their article (what does this mean?). If published, this will include your full peer review and any attached files.

Reviewer #1: No

Reviewer #2: **Yes: **Janine M. Ziermann-Canabarro

Reviewer #3: No

---

## [Editor Report · Decision Letter 1]

5 Mar 2024

Dear Dr Comai

We are pleased to inform you that your manuscript entitled "Interplay between Pitx2 and Pax7 temporally governs specification of extraocular muscle progenitors" has been editorially accepted for publication in PLOS Genetics. Congratulations!

Yours sincerely,

Peter S Zammit

Guest Editor

PLOS Genetics

Gregory Barsh

Editor-in-Chief

PLOS Genetics

Comments from the guest editor:

Some minor suggestions to enhance ‘readability’.

• As shown by the confusion of reviewer 1, labeling could be clearer on figures with more explanation in figure legends. For example, in Figure 2 the labeling of G/+ and G/G for the X-axis does not immediately indicate what is being measured. Expanded labels would be better and/or a key in the figure legend. Supplemental Table 1 is useful, and ‘codes’ such as ‘G’ could also be listed here.

• Area in Fig 2C should be properly indicated using mu, a capital ‘M’ and superscripted 2.

• Line 93: should be Fig S1?

• Fig 5: scale bar should not refer to panel A

• Would mention that efforts were made to generate Pax7CreERT2; Pitx2flox KO mice induced at E11.5+E12.5 to be analyzed postnatally but that no living pups were obtained even when progesterone was co-administered.

• Figure 7: two panels labelled J, summary panel should be K. Could make Fig. 7K alone into figure 8 as it presents a useful summary of a lot of data and would benefit therefore, from being at a reasonable size. Would refer to it more in the discussion.

• Due to the other data on EOMs in the literature, would refer more to specific species e.g. the rabbit data, in the discussion.

**Data Deposition**

http://datadryad.org/submit?journalID=pgenetics&manu=PGENETICS-D-23-00956R1

**Press Queries**

---

## [Editor Report · Acceptance letter]

15 Apr 2024

PGENETICS-D-23-00956R1 

Interplay between Pitx2 and Pax7 temporally governs specification of extraocular muscle progenitors 

Dear Dr Comai, 

We are pleased to inform you that your manuscript entitled "Interplay between Pitx2 and Pax7 temporally governs specification of extraocular muscle progenitors" has been formally accepted for publication in PLOS Genetics! Your manuscript is now with our production department and you will be notified of the publication date in due course.

With kind regards,

Anita Estes

PLOS Genetics

On behalf of:
